# Photo-produced aromatic compounds stimulate microbial degradation of dissolved organic carbon in thermokarst lakes

Jie Hu[1,2], Luyao Kang[1,2], Ziliang Li[1,2], Xuehui Feng[1,2], Caifan Liang [1,2], Zan Wu[1], Wei Zhou[1,2], Xuning Liu[1,2], Yuanhe Yang [1,2] & Leiyi Chen [1] ✉

Photochemical and biological degradation of dissolved organic carbon (DOC) and their interactions jointly contribute to the carbon dioxide released from surface waters in permafrost regions. However, the mechanisms that govern the coupled photochemical and biological degradation of DOC are still poorly understood in thermokarst lakes. Here, by combining Fourier transform ion cyclotron resonance mass spectrometry and microbial high-throughput sequencing, we conducted a sunlight and microbial degradation experiment using water samples collected from 10 thermokarst lakes along a 1100-km permafrost transect. We demonstrate that the enhancement of sunlight on DOC biodegradation is not associated with the low molecular weight aliphatics produced by sunlight, but driven by the photo-produced aromatics. This aromatic compound-driven acceleration of biodegradation may be attributed to the potential high abilities of the microbes to decompose complex compounds in thermokarst lakes. These findings highlight the importance of aromatics in regulating the sunlight effects on DOC biodegradation in permafrost-affected lakes.

Permafrost soils store considerable quantities of organic carbon (C)[1], almost twice the amount of C already present in the atmosphere[2,3]. Rapid climate warming has led to a widespread permafrost thaw and swift conversion of permafrost-stored C into atmosphere C dioxide ($CO_2$)[4,5], which may significantly amplify climate change. Thermokarst lake, one typical thermokarst landform covering a certain area of the thermokarst area[6], is a hotspot of C emission in permafrost region[7]. It has been estimated that the C release from thermokarst lakes will far outweigh that from gradual permafrost thaw in terms of active layer deepening by 2300[8,9]. Dissolved organic C (DOC) is an important source of this C release, and its decomposition was jointly controlled by sunlight and microbial processing[10,11]. Particularly, sunlight may act as an amplifier during DOC degradation[12], since it could not only directly convert DOC to $CO_2$ via photo-mineralization[13,14], but also affect microbial decomposition by modifying the chemical composition of DOC[15,16]. Therefore, knowledge concerning the underlying

mechanism of sunlight effect on the microbial degradation of DOC in thermokarst lakes is crucial for the accurate projection of C release and its associated feedback to climate warming in the permafrost region.

Because of the vital role of sunlight in governing DOC fate, the responses of DOC biodegradation to sunlight have garnered long-term attention[17,18]. It has been demonstrated that sunlight can exert contrasting effects on how DOC is processed by microbial communities[19], which depends on the compounds that microbes and light may degrade[10]. It has traditionally been assumed that small (low molecular weight) and aliphatic-like DOC compounds were more preferentially degraded by microbes than aromatic-like compounds (hereafter "traditional view")[17]. Since sunlight is deemed to convert the larger (high molecular weight) and more aromatic-like compounds to smaller compounds[20,21], sunlight exposure would thus enhance DOC biodegradation[17,18]. Nevertheless, a growing volume of evidence challenged this view by demonstrating that terrestrial DOC with high

[1]State Key Laboratory of Vegetation and Environmental Change, Institute of Botany, Chinese Academy of Sciences, Beijing 100093, China. [2]University of Chinese Academy of Sciences, Beijing 100049, China. ✉e-mail: chenly@ibcas.ac.cn

aromaticity may be more important in fueling microbial respiration than the small pools of labile aliphatic-like compounds (hereafter "emerging view")[22–24]. Therefore, sunlight may also inhibit microbial degradation by competing with microbes for these aromatic compounds[25], or promote microbial processes by converting high molecular weight aromatic compounds into more biodegradable aromatic compounds[15]. Despite all the extensive empirical progress in arctic and temperate ecosystems[13,17,19,25], the evidence for the biological and photochemical degradation of DOC in thermokarst lakes is still lacking. It remains unknown which of the two alternative views would apply in this unique ecosystem. It was reported that microbes in the permafrost DOC trended to degrade more aromatic and oxidized DOC along permafrost thawing gradients on the plateau[26]. Meanwhile, the dominance of terrestrial-derived aromatic compounds in thermokarst lakes would increase with permafrost thawing[19,27]. Therefore, aromatic compounds rather than low molecular weight and aliphatic-like DOC compounds were expected to stimulate microbial respiration of DOC in thermokarst lakes. Here, we hypothesized that consistent with the emerging view[15,25,28], sunlight may promote microbial degradation by converting high molecular weight aromatic-like DOC into more biodegradable aromatic compounds in thermokarst lakes.

The Tibetan Plateau is the largest alpine permafrost region in the world and contains approximately 161,300 thermokarst lakes[29,30]. Rapid climate warming and consequent widespread permafrost thaw have induced a surge in the number of thermokarst lakes across the plateau over the past four decades[30]. However, studies concerning

the sunlight effect on microbial degradation of DOC in thermokarst lakes are still lacking on the plateau. To fill this knowledge gap, we collected water samples from 10 thermokarst lakes across a 1100-km transect on the plateau. Afterwards, we conducted an ultraviolet (UV) sunlight and microbial degradation experiment in combination with UV-visible (UV-Vis) spectrophotometry, fluorescence spectroscopy, Fourier transform ion cyclotron resonance mass spectrometry (FT-ICR MS), and microbial high-throughput sequencing to uncover the mechanisms underlying the response of DOC biodegradation to sunlight. Particularly, FT-ICR MS is an ultrahigh-resolution analytical technique that can provide unparalleled insight into the molecular composition of dissolved organic matter (DOM)[31,32]. Based on the molecular formulas derived from FT-ICR MS, four types of functional compounds with declining photoreactivity and aromaticity, including combustion-derived polycyclic aromatics (CA), vascular plant-derived polyphenols (Pol.), highly unsaturated and phenolic compounds (Uns.) and aliphatic compounds (Ali.) were identified[32]. The response ratio of microbial $CO_2$ respiration to sunlight (hereafter referred to as $CO_2$-RR), defined as the natural logarithm of the ratio between the sunlight treatment and dark control, was used to facilitate the comparison of the responses of DOC biodegradation to sunlight among lakes. Our results demonstrate that the enhancement of sunlight on DOC biodegradation was induced by photo-produced, aromatic-like compounds rather than low molecular weight, aliphatic-like compounds. This aromatic compound-driven acceleration of microbial DOC degradation could be possibly attributed to

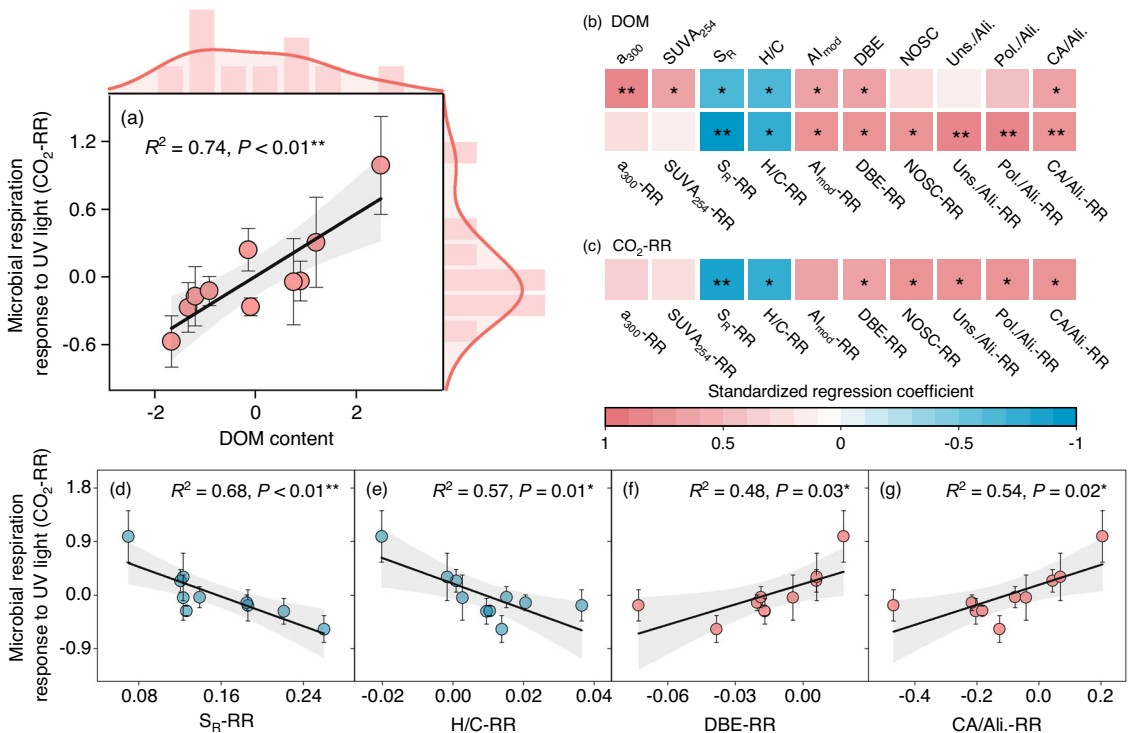

**Fig. 1 | Relationships of microbial respiration response to sunlight with dissolved organic matter (DOM) content and its chemistry. a** Relationships of response ratio (RR, the ratio of variable between sunlight treatment and dark control [Variable_sunlight/Variable_dark]) of microbial respiration ($CO_2$-RR) with DOM content, which is represented as the first component of principal components analysis for DOC and dissolved organic nitrogen content. Marginal histograms and density plots show the frequency distribution of $CO_2$-RR and DOM content across the ten lakes. **b–c** Standardized regression coefficients of the factors of DOC chemistry and their RRs with DOM content (**b**) and $CO_2$-RR (**c**), with the color indicating the strength and sign of the relationship. **d–g** Relationships of $CO_2$-RR with the response ratios of DOC chemistry. The solid line and grey area in subfigures (**a, d–g**) represent the linear regression lines and the 95% confidence interval, respectively. Dots with bars indicate means ± standard error (SE) ($n = 3$). The two-sided statistical tests indicate significant effects by *$P < 0.05$, **$P < 0.01$. a_{300}, the Naperian absorption coefficient at 300 nm; SUVA_{254}, the absorbance at 254 nm divided by DOC concentration; $S_R$, slope coefficient ratio correlated to DOC molecular weight; H/C, the ratio of the number of hydrogen atoms to the number of carbon atoms; AI_{mod}, modified aromaticity index; DBE, the double bond equivalence; NOSC, the nominal oxidation state of carbon; Uns./Ali., the ratio of highly unsaturated and phenolic compounds to aliphatic compounds; Pol./Ali., the ratio of vascular plant-derived polyphenols to aliphatic compounds; CA/Ali., the ratio of combustion-derived polycyclic aromatics to aliphatic compounds.

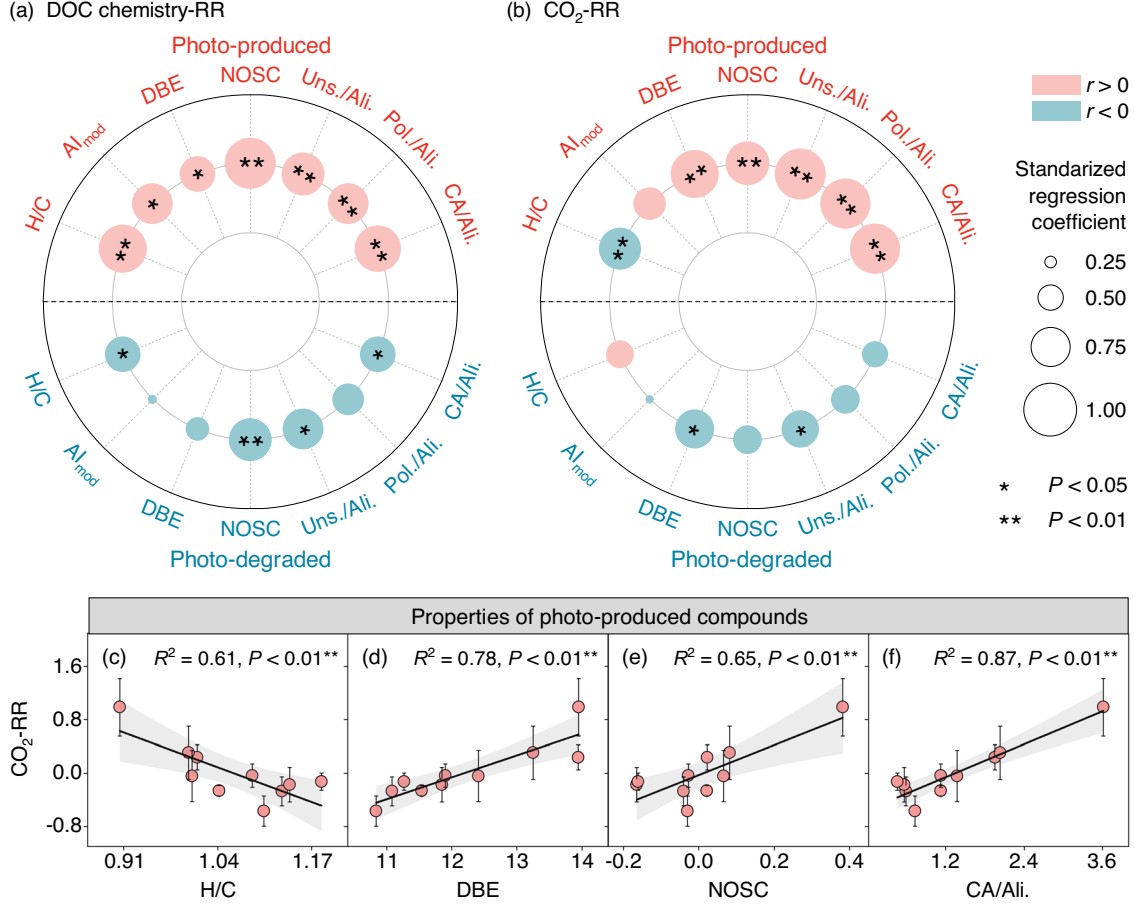

**Fig. 2 | Relationships of response ratios (RR) of dissolved organic carbon (DOC) chemistry and microbial respiration (CO$_2$-RR) with the properties of photo-produced and photo-degraded compounds. a, b** Standardized regression coefficients of the properties of photo-produced and photo-degraded compounds with their perspective RRs (**a**) and CO$_2$-RR (**b**), with the bubble size and color indicating the strength and sign of the relationship. **c–f** Correlations of CO$_2$-RR with the DOC chemistry of photo-produced compounds. The solid line and grey area represent the linear regression lines and the 95% confidence interval, respectively. Dots with bars in subfigures (**c–f**) indicate means ± standard error (SE) ($n = 3$). The two-sided statistical tests indicate significant effects by *$P < 0.05$, **$P < 0.01$. H/C, the ratio of the number of hydrogen atoms to the number of carbon atoms; AI$_{mod}$, modified aromaticity index; DBE, the double bond equivalence; NOSC, the nominal oxidation state of carbon; Uns./Ali., the ratio of highly unsaturated and phenolic compounds to aliphatic compounds; Pol./Ali., the ratio of vascular plant-derived polyphenols to aliphatic compounds; CA/Ali., the ratio of combustion-derived polycyclic aromatics to aliphatic compounds.

the potential high abilities of the microbes to decompose complex compounds in thermokarst lakes.

## Results and discussion
### Sunlight effect on DOC biodegradation varies with regional DOM content

A UV sunlight and subsequent microbial degradation experiment in two temperatures (i.e., 10 °C, 20 °C; See Methods for details) were conducted to explore the coupled photochemical and biological degradation of DOC. No significant interactions between sunlight and temperature (S × T) were found for microbial respiration, demonstrating that the sunlight effect on DOC biodegradation was independent of the temperature ($P = 0.49$, Supplementary Table 1). Nevertheless, the sunlight effect on microbial DOC degradation exhibited substantial variability across the 10 thermokarst lakes, ranging from −0.57 to 0.99 (Supplementary Fig. 1), which was supported by the significant interaction between the sunlight and lake location (S × L) ($P < 0.05$, Supplementary Table 1). Such regional variation in CO$_2$-RR was independent of DOC chemistry (Supplementary Fig. 2), but positively associated with DOM content (i.e., the first component (PC1) of principal components analysis (PCA) for DOC and dissolved organic nitrogen (DON) content) ($R^2 = 0.74$, $P < 0.01$; Fig. 1a). Meanwhile, the DOM content was positively correlated with contents of

chromophoric dissolved organic matter (CDOM, a$_{300}$), specific UV absorbance (SUVA$_{254}$), modified aromaticity index (AI$_{mod}$), double bond equivalence (DBE) and the ratio of CA to Ali., but negatively related to slope coefficient ratio (S$_R$) and H/C ratio (Fig. 1b), indicating that the thaw lakes with high DOM content generally have more large (i.e., high molecular weight, low S$_R$), terrestrially derived aromatic-like compounds (i.e., more CDOM, higher a$_{300}$; higher SUVA$_{254}$ and AI$_{mod}$), but less small (i.e., low molecular weight, high S$_R$), aliphatic-like (i.e., higher H/C elemental ratios) compounds. This relationship was also supported by the observations in arctic thermokarst-impacted water[19], implying that the thermokarst lakes with higher DOM content may contain more terrestrial-derived aromatic compounds[23].

### Boost of light on DOC biodegradation via photo-produced aromatics

To uncover the drivers of the sunlight effect on DOC biodegradation, we further explored the responses of biotic and abiotic factors to sunlight exposure. Firstly, given that most microbes were eliminated by filtering the water samples through 0.22 μm filters before the sunlight experiment (Supplementary Table 2), sunlight exposure would barely affect the microbes. Additionally, we assumed that the reactive oxygen species (ROS) could largely decay during the 12-h placement after the sunlight experiment[25], and its impact on biodegradation was

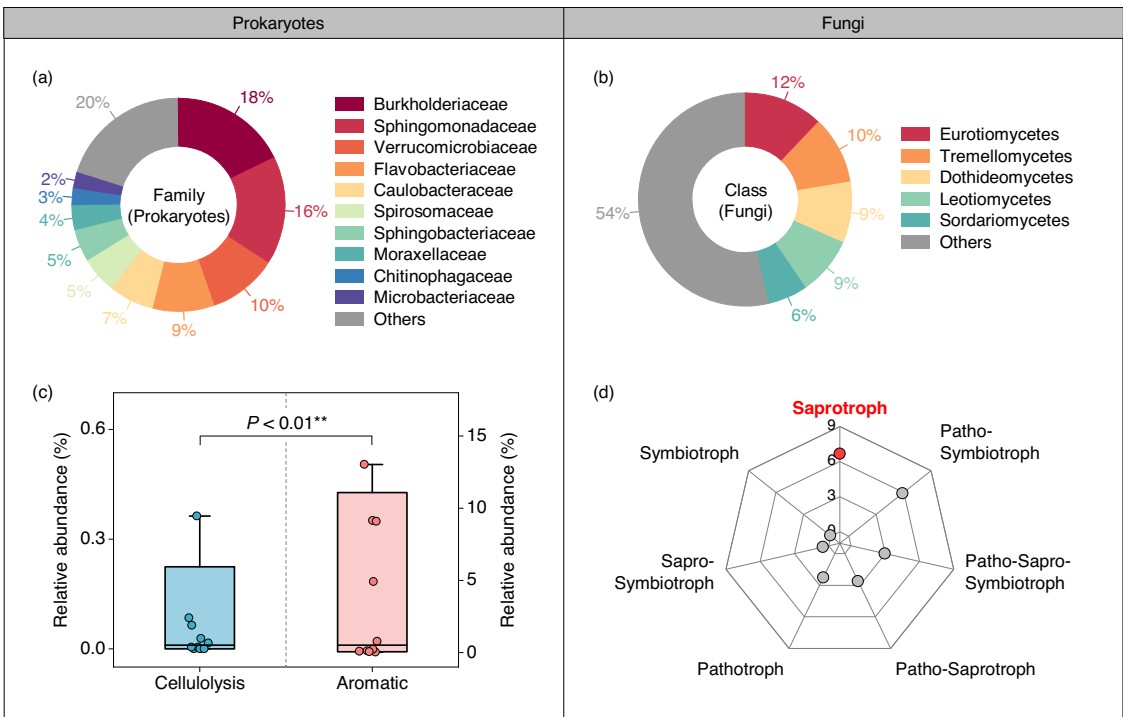

**Fig. 3 | Taxonomic composition and function of microbial communities across the thermokarst lakes.** The taxonomic composition of prokaryotic (**a**) and fungal (**b**) community at the Family and Class level, respectively. **c, d** The relative abundance (%) of prokaryotic functional groups involving in degrading cellulose and aromatic compounds based on the FAPROTAX analysis (**c**) and fungal trophic groups with FUNGuild analysis (**d**). The whiskers illustrate the 1st and 99th percentiles, and the ends of the boxes represent the 10th and 90th quartiles. The horizontal line inside each box shows the median. Statistical significance between the relative abundance of prokaryotes degrading cellulose and aromatic compounds was determined with a two-tailed Wilcoxon rank-sum test ($n = 10$). **$P < 0.01$. Patho: Pathotroph, Sapro: Saprotroph.

also minimal in our study (See Methods for details). Hence, we concentrated solely on the changes in water properties after sunlight treatment. We found that although sunlight did not affect the content of DOC, dissolved inorganic carbon (DIC), DON and pH value of the water samples (Supplementary Fig. 3a–d), it altered DOC chemistry considerably. Generally, sunlight exposure significantly reduced the amount of CDOM and aromatic compounds (i.e., decreased $a_{300}$ and $SUVA_{254}$), resulting in the decline in DOC molecular weight (i.e., increased $S_R$) (Supplementary Fig. 4a–c). Nevertheless, the sunlight effect on DOC chemistry varied considerably across thermokarst lakes, indicated by the significant interaction of sunlight and lake location (S × L) (Supplementary Fig. 4a–c) and their large variations across the 10 lakes (i.e., H/C, DBE, and CA/Ali. ratio, Supplementary Fig. 4d–f), suggesting that the distinct photochemical alteration of DOC chemistry across the lakes might be the potential driver of the spatial variation of sunlight effect on DOC biodegradation.

To examine the potential linkage between DOC biodegradation and changes in DOC chemistry after sunlight exposure, we further calculated the response ratios (RRs) of DOC chemical properties to sunlight and established their links to DOM content and $CO_2$-RR. Our results showed that after sunlight exposure, the lakes with higher DOM generally produced less small and aliphatic-like compounds, indicated by the negative associations of DOM content with small (i.e., low molecular weight, high $S_R$) and aliphatic (i.e., high H/C) DOC (Fig. 1b). Furthermore, sunlight exposure even led to a certain increase in aromatic-like compounds in thermokarst lakes with high DOM (Fig. 1g). More interestingly, in contrast to the traditional view[17,18], the enhancement of sunlight on DOC biodegradation was not driven by the photo-produced aliphatic-like compounds, but associated with the photo-produced aromatic-like compounds, as evidenced by the $CO_2$-RR decreasing with smaller (i.e., lower molecular weight, higher $S_R$) and more aliphatic (i.e., higher H/C) DOC, but increasing with more unsaturated (i.e., higher DBE), more oxidized (i.e., higher the nominal oxidation state of carbon (NOSC)) DOC and higher proportions of the aromatic-like compound to the aliphatic-like compound (i.e., higher ratios of Uns./Ali., Pol./Ali., CA/Ali.) after sunlight exposure (Fig. 1c–g).

Given that the sunlight effect on DOC chemical properties was the consequence of the photo-production and photo-removal of DOC, we further adopted FT-ICR MS technology to reveal the molecular composition of photo-produced and photo-degraded compounds. On the whole, the sunlight effect on DOC chemistry was more attributed to photo-produced compounds than photo-degraded compounds (Fig. 2a). Moreover, compared with lakes with low DOM content, light exposure produced more aromatic-like compounds in lakes with high DOM content, represented by the positive correlation of the DOM content with aromaticity (higher $AI_{mod}$), unsaturation (higher DBE), oxidability (higher NOSC), and the proportions of the aromatic-like compound to the aliphatic-like compound (higher CA/Ali. ratio) (Supplementary Fig. 5d–g) and the negative relationship with the lability (i.e., higher H/C) and the proportion of aliphatic-like compounds in photo-produced compounds (Supplementary Fig. 5b–c). This may be related to the fact that these lakes originally contained more aromatic compounds (Fig. 1b). Furthermore, these photo-produced unsaturated (i.e., higher DBE), oxidized (i.e., higher NOSC) and aromatic-like compounds (i.e., the number of Uns., Pol., CA, and the CA/Ali. Ratio) were found to promote DOC biodegradation indicated by their positive associations with $CO_2$-RR (Fig. 2b–f; Supplementary Fig. 6b–d). By contrast, photo-produced aliphatic compounds were observed to inhibit DOC biodegradation indicated by its negative association with $CO_2$-RR (Supplementary Fig. 6a). In accordance with our findings, a recent study conducted in the temperate stream also found that high molecular weight aromatic-like compounds could be converted into low molecular weight aromatics after sunlight exposure, further promoting the subsequent microbial degradation[15]. This

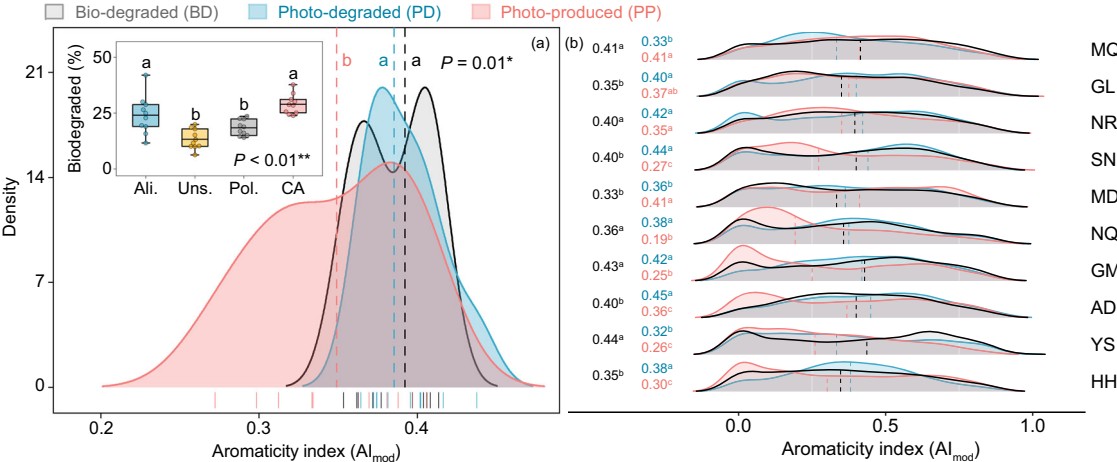

**Fig. 4 | Chemical properties of photo-produced (PP), photo-degraded (PD), and bio-degraded (BD) compounds.** The average modified aromaticity index ($AI_{mod}$) of PP, PD, and BD compounds of 10 lakes (**a**) and the $AI_{mod}$ of each lake (**b**). The sampling sites of the ten lakes are Maqên County (MQ), Golog Tibetan Autonomous Prefecture (GL), Nyainrong County (NR), Seni District (SN), Madoi County (MD), Nagqu (NQ), Golmud (GM), Amdo County (AD), Yushu (YS) and Heihe River (HH). Insert box plots in panel (**a**) display the relative abundance of four types of BD compounds (i.e., Ali., aliphatic compounds, $2.0 \geq H/C \geq 1.5$; Uns., highly unsaturated and phenolic compounds, $AI_{mod} \leq 0.50$, $H/C < 1.5$; Pol., vascular plant-derived polyphenols, $0.66 \geq AI_{mod} > 0.50$; CA, combustion-derived polycyclic aromatics, $AI_{mod} > 0.66$). The whiskers illustrate the 1st and 99th percentiles, and the ends of

the boxes represent the 25th and 75th quartiles (interquartile range). The horizontal lines inside each box show the mean ($n = 10$). Statistical significance was assessed using a two-tailed one-way ANOVA with significant effects indicated as $*P < 0.05$, $**P < 0.01$. Different lowercase letters indicate significant differences among respective groups based on two-sided tests for multiple comparisons by FDR corrections ($P < 0.05$). The black, blue, and pink number in panel (**b**) represents the median of the $AI_{mod}$ of BD, PD, and PP compounds, respectively. A two-tailed Wilcoxon rank-sum test was used to assess the statistical significance in panel (**b**) with different letters indicating significant differences among the respective groups ($P < 0.05$).

could be ascribed to the similar function of sunlight to extracellular enzymes in degrading high molecular weight DOC[28], which can reduce the energy required to break chemical bonds and thus promote the decomposition of the complex compounds[33]. Taken together, in support of the emerging view[15,25,28], but contradicting the traditional view[17,18,34] in arctic or temperate ecosystems, our results demonstrated that the promotion of microbial degradation by sunlight increased with the aromaticity of photo-produced compounds in thermokarst lakes.

### Microbial function and their linkage to the sunlight effect

An interesting question arises as to why photo-produced aromatic-like compounds, rather than low molecular weight aliphatic-like compounds, promote DOC biodegradation. It has been proposed that microbes may have the intrinsic capability to degrade the most abundant DOC in the environment regardless of their molecular weight and aromaticity[25]. To test this possibility, we first used high-throughput sequencing (prokaryotic 16S rRNA gene, fungal ITS rRNA gene) to elucidate the overall profile of the microbial composition in thermokarst lakes. We found that Burkholderiaceae were the most dominant family in prokaryotes (i.e., bacteria and archaea), which comprised 18% of total sequence reads, followed by Sphingomonadaceae (16%), Verrucomicrobiaceae (10%), and Flavobacteriaceae (9%) (Fig. 3a). Similarly, Eurotiomycetes and Tremellomycetes were the dominant groups at the class level in fungi, accounting for 12% and 10% respectively (Fig. 3b).

To further reveal the potential functions of the microbial community in the thermokarst lakes, FAPROTAX[35] and FUNGuild[36] were used to predict prokaryotic and fungal functions based on Operational Taxonomic Unit (OTU) taxonomic information, respectively. The analysis showed that the relative abundances of prokaryotes involved in aromatic compounds degradation were significantly higher than those associated with cellulose degradation in the thermokarst lakes (Fig. 3c). Likewise, saprophytic fungi were the most abundant functional groups (Fig. 3d). This fungal group has been reported to be strongly associated with aromatic compound degradation[37]. All of

these findings were in accord with a recent study indicating that the taxa and copiotrophs in Tibetan lakes were actively involved in the degradation of terrestrially-derived DOC[38]. Moreover, the significantly positive linkage between the enhancement of sunlight on DOC biodegradation and fungal richness ($R^2 = 0.94$, $P < 0.01$; Supplementary Fig. 7) further demonstrated the critical role of microbial community structure in driving the sunlight effect on DOC biodegradation.

Besides these potential functional predictions, the properties of biodegraded compounds were also evaluated to verify microbial functions in degrading DOC. Our analysis showed that the chemistry of compounds consumed by microbes (i.e., bio-degraded DOC) was generally more comparable to the compounds removed by sunlight (i.e., photo-degraded DOC), but had higher aromaticity (i.e., $AI_{mod}$) and higher unsaturation (i.e., DBE) than those compounds produced by sunlight (i.e., photo-produced DOC) (Fig. 4; Supplementary Fig. 8). These results inferred that the ability of microbes to degrade the recalcitrant DOC with high aromaticity and unsaturation was similar to the photodegradation in the thermokarst lakes. Likewise, we also found the proportion of combustion-derived polycyclic aromatics (CA) degraded by microbes was much higher than that of highly unsaturated and phenolic compounds (Uns.) and vascular plant-derived polyphenols (Pol.), and was similar to the aliphatic compounds (Ali.) (Fig. 4a insert box plots), confirming that microbes in the thermokarst lakes tend to degrade these highly aromatic-like compounds.

By linking the microbial properties to the changes in DOC molecular composition after sunlight exposure, our study illustrated that the sunlight effect on DOC biodegradation at the regional scale depended on whether light produced or removed the compounds that fueled native microbes. For instance, for lakes (i.e., MQ, GL, NR) with positive sunlight effect on DOC biodegradation, there was a strong overlap in the aromaticity ($AI_{mod}$) of photo-produced compounds and bio-degraded compounds (Fig. 4b). Thus, light exposure was to produce more of the similar aromatic-like compounds that their microbes consumed in these lakes. In contrast, in the remaining lakes where sunlight inhibited microbial degradation, light exposure removed the compounds that were consumed by their microbes in the dark, as

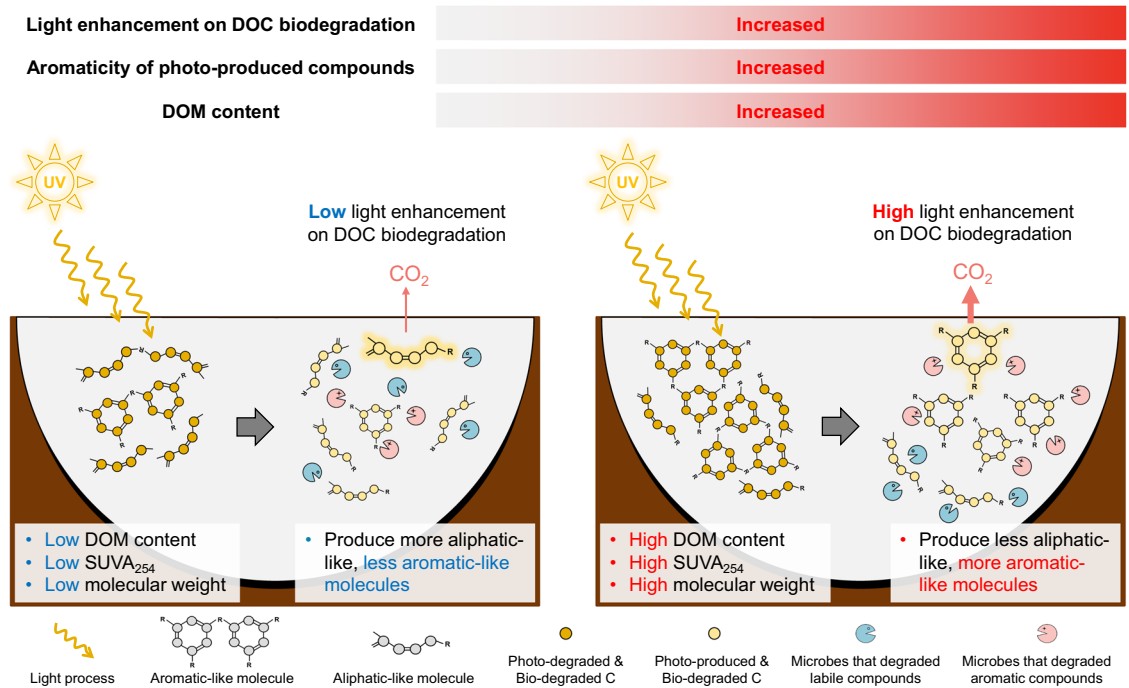

**Fig. 5 | Conceptual diagram of the sunlight effect on microbial degradation of dissolved organic carbon (DOC) in thermokarst lakes.** With the increase of DOM content (represented as the first component of principal components analysis for DOC and dissolved organic nitrogen content), more small aromatic-like compounds will be produced by sunlight exposure. Given the high abilities of microbes to decompose complex compounds in thermokarst lakes, lakes with more increase in aromatic-like compounds after UV exposure will exhibit larger enhancement of sunlight on subsequent microbial degradation of DOC. The conceptual diagram was created using Microsoft PowerPoint 2016.

evidenced by the strong overlap in the aromaticity ($AI_{mod}$) of photo-degraded compounds and bio-degraded compounds (Fig. 4b). This finding was consistent with the emerging view derived from site-level soil DOC incubations[25], which jointly demonstrated the overlap between light-altering and microbial degraded compounds determines the sunlight effect on microbial degradation.

In summary, consistent with the emergent view[15,25,28] but contrasted with the traditional view[17,18,34] found in arctic and temperate ecosystems, our results demonstrated that the enhancement of sunlight on DOC biodegradation in thermokarst lakes across the Tibetan Plateau was governed by the more oxidized and aromatic-like compounds rather than the low molecular weight, aliphatic-like compounds produced by sunlight (Fig. 5). Such aromatic-compound drove DOC biodegradation may be ascribed to the high abilities of microbes to decompose complex compounds in thermokarst lakes. These findings have important implications for understanding C release from thermokarst lakes in the context of global warming. First, the aromatic compound-induced acceleration of microbial DOC degradation indicates that not only the small aliphatic but also the aromatic-like compounds produced by sunlight can fuel subsequent microbial decomposition. It is thus necessary to consider the overlap between light-altering and microbial-degraded compounds for better predicting the $CO_2$ release from thermokarst lakes. In addition, the close link positive association of sunlight effects on DOC biodegradation with DOM content suggested that the lakes with higher DOM content and more terrestrially derived aromatic DOC could produce more aromatic-like compounds after sunlight exposure, thereby significantly contributing to the $CO_2$ release from thermokarst lakes. Given that these lakes are generally located in swamp meadows and alpine meadows receiving high plant C input, the higher enhancement of UV sunlight on DOC biodegradation in these lakes thus can partially offset the vegetation C fixation, amplifying the positive C-climate feedback in these permafrost regions. Furthermore, compared to the shallow arctic lakes[12], the photon flux in the thermokarst lakes on the

Tibetan Plateau is generally higher (Supplementary Fig. 9), despite the similar content of CDOM in the two regions ($a_{305}$: 7.37-60.57 m$^{-1}$ in this study vs 1.05-77.46 m$^{-1}$ for arctic lakes) (Supplementary Table 3 and[12]). Given the rate of light absorption by CDOM ($Q_{a,\lambda}$) depends on the both light available and the content of CDOM, the high photon flux infers the high rates of photochemical process on the Tibetan Plateau. Specifically, the rate of light absorption by CDOM was estimated to range from 95.10 to 153.21 μmol photon m$^{-2}$ s$^{-1}$ in the 280–400 nm across the 10 lakes (Supplementary Fig. 10; see Supplementary Methods for details), inferring that DOC photodegradation could be an important source of $CO_2$ from these thermokarst lakes across the Tibetan Plateau.

## Methods

### Study area and water sampling

During July to August from 2019 to 2020, we collected water samples from 196 thermokarst lakes at 48 sites along a 1100-km permafrost transect across the Tibetan Plateau. Before collecting water samples, water conductivity, salinity and dissolved oxygen were measured at the three locations from the shore to the centre of each lake using a Professional Plus (Pro Plus) multiparameter instrument (YSI, Yellow Springs, USA). Afterwards, three equal volumes of the surface water samples (0–20 cm) were collected at the corresponding locations and mixed as one composite sample. Then, 3 L water samples were filtered (GF/F filters, Whatman) into acid-washed amber high-density polyethylene (HDPE) bottles for subsequent chemistry analysis and dissolved organic carbon (DOC) degradation experiment. Considering the high experimental cost, we selected 10 representative thermokarst lakes from 48 sites to perform the sunlight and microbial degradation experiment. These thermokarst lakes encompass a wide precipitation gradient with mean annual precipitation spanning from 307 to 497 mm and also cover a wide range of DOC quantities and composition variations across the region, i.e., DOC (5.69–16.76 mg L$^{-1}$), dissolved organic nitrogen (DON, 0.33–1.35 mg L$^{-1}$), chromophoric dissolved organic matter (CDOM, $a_{300}$, 8.44–65.87 m$^{-1}$) and aromatic

compounds (SUVA$_{254}$, 1.40–3.49 L mg C$^{-1}$ m$^{-1}$) (Supplementary Table 3). Despite these variations, the biological index (BIX) values of DOC ranged from 0.5 to 0.8, inferring the DOC in these thermokarst lakes were mainly terrestrial origins (Supplementary Table 3)[39]. In addition, these thermokarst lakes are generally located in swamp meadows and alpine meadows with limited vegetation growing in the lakes (Supplementary Fig. 11). Therefore, no vegetation shading will affect the importance of photo-bio degradation in these lakes. To obtain the microbial inocula of the ten lakes for the DOC biodegradation experiment, we also filtered 100 mL of fresh water through GF/C filters[19]. All these filtrations were conducted within 24 h after sampling and transported to the lab at −20 °C until laboratory analysis. Although this freezing temperature was generally considered adequate to store samples for microbial analysis[40], this storage would inevitably reduce microbial activities. Therefore, to minimize this potential effect, pre-incubation at 20 °C was conducted for the inocula to activate the microbe and avoid pulses in microbial activities induced by changing temperature before the microbial degradation experiment[41–43].

## Photobiological experimental design

We conducted an ultraviolet (UV) light and biodegradation experiment in the lab to explore the light effect on DOC biodegradation across the thermokarst lakes. The water samples were frozen until the UV light experiment was set up within 4 to 6 weeks. All the samples were thawed before the start of the UV light experiment in the dark at room temperature (20 °C) for around 10 hours. Then, the UV light experiment (i.e., both the UV exposure treatment and dark control) was conducted at room temperature. To remove microbes during the UV degradation, we isolated the GF/F filtered water samples further with 0.22 μm filters (Sterivex GP 0.22, Millipore) to generate triplicate samples for each thermokarst lake. This pore size of the filters was widely used in previous studies[19,44] since it can eliminate most of the microbial populations (e.g., prokaryotes and fungi) in the water (Supplementary Table 2). Afterwards, the filtered samples were placed in UV-transparent Whirl-Pak bags (Nasco, Inc.) and exposed to stimulated UV light (40-W, UVA-340 tubes, spectral range: 315–400 nm) for 21 h alongside triplicate foil-wrapped Whirl-Pak bags as dark control. In total, we set up 60 Whirl-Pak bags (10 lakes × 2 light treatments × 3 replicates) in the photo-degradation experiment.

After exposure to UV light, subsamples were analyzed for DOC, nutrient content, CDOM, fluorescent dissolved organic matter (FDOM), and DOC molecular composition. Given the high cost of Fourier transform ion cyclotron resonance mass spectrometry (FT-ICR MS), DOC molecular composition was only assessed by using the composite sample of the three replicates for each thermokarst lake after UV exposure. Meanwhile, the remaining water sample for each light-exposed or dark control replicate was transferred into a 50 mL amber jar with an airtight lid after 12-hour placement, during which reactive oxygen species (ROS) was assumed to be largely quenched and would exert limited impact on microbes[25]. Afterwards, the inoculum (i.e., GF/C filtered water from the lake corresponding to the light-exposed and dark control) of approximately 6% of the sample volume was inoculated, which was pre-incubated (20 °C) to activate microbes and avoid pulses in microbial activities induced by changing temperature[41–43]. This combination of freezing storage and pre-incubation has been widely used in previous large-scale soil incubations[41–43]. It should be noted that despite this pre-incubation may cause potential "bottle effects" on microbial community composition[45–47], it would not influence the treatment comparisons because the water samples in the light-exposed treatment and dark control received the same inoculum from the corresponding lake. Additionally, water samples that received equal volumes of sterile water were set as blank to eliminate the shift in headspace carbon dioxide (CO$_2$) remaining in the bottle after flushing with CO$_2$-free air

during the incubation. After pre-incubation, all the water samples were flushed with CO$_2$-free air for 10 min to homogenize the initial headspace CO$_2$ concentration in the jar (Supplementary Fig. 12)[48,49]. Then, all bottles were incubated for 28 days in a dark environment[50] at 10 °C and 20 °C. To be specific, 10 °C was chosen because the daytime mean lake surface water temperature (LSWT) during the growing seasons (i.e., from May to October) in 2001–2015 was estimated at approximately 12 °C across the Tibetan Plateau[51]. Meanwhile, room temperature at 20 °C was recommended to perform DOC incubation in permafrost soils and aquatic systems, as this is the most common, relatively easy to maintain, and allows comparison among studies[50]. The bottles were shaken every day during the incubation to ensure oxygen supply[50].

Microbial CO$_2$ productions were calculated as the changes in headspace CO$_2$ in the jar over time, which were measured on days 0, 7 and 28 by adding 1 mL of H$_3$PO$_4$ (40%) to convert dissolved inorganic carbon (DIC) in water to headspace CO$_2$[52]. Thus, microbial CO$_2$ production after 7- or 28-day incubation was calculated as the difference between the cumulative CO$_2$ productions measured on day 7 or 28 and that measured on day 0, which was the background DIC content in the water sample. Consequently, a total of 384 microcosms (60 Whirl-Pak bags × 2 temperatures × 3 time periods + 24 blanks) were set up to measure microbial CO$_2$ production. In addition, to explore the DOC molecular composition of biodegraded compounds, the composite samples for each thermokarst lake after UV exposure was also divided into two aliquots and incubated at 10 °C and 20 °C with inoculum synchronously. After 28-day incubation, the effects of biodegradation on DOC molecular composition were analyzed by FT-ICR MS.

## Water chemistry, CDOM and FDOM analysis

Water chemistry, CDOM and FDOM content were analyzed for both background GF/F filtered samples and water samples after the UV experiment. Of them, DOC and total dissolved nitrogen (TDN) were analyzed by a multi-N/C 3100 analyzer (Analytik Jena, Germany). NH$_4^+$-N and NO$_3^-$-N were quantified by a flow injection analyzer (SEAL Analytical Ltd., Southampton, UK). DON was calculated by subtracting dissolved inorganic nitrogen (DIN, total of NH$_4^+$-N and NO$_3^-$-N) from TDN. Notably, since the concentration of NO$_3^-$-N was below the detection limit, this variable was not included in the analysis. The pH value was determined with a pH electrode (PB-10, Sartorius, Germany).

The optical properties of water samples were assessed by UV and Fluorescence Spectroscopy. Of them, UV absorbances at 200–600 nm were measured by UV-visible (UV-Vis) spectrophotometer (Lambda35, Perkin Elmer Inc., Waltham, USA)[53]. The Naperian absorption coefficient at 300 nm (a$_{300}$), a proxy of the amount of CDOM, was then calculated by multiplying the absorbance at 300 nm by 2.303 and dividing it by the optical path length[19]. SUVA$_{254}$ (L mg C$^{-1}$ m$^{-1}$), a proxy of DOC aromaticity, was calculated as absorbance at 254 nm divided by path length (m) and DOC concentration (mg C L$^{-1}$)[54]. Spectral slope ratio (S$_R$), the ratio of the spectral slope S$_{275-295}$ to S$_{350-400}$, was calculated by linear regression from the logarithm of absorbance between 275–295 nm and 350–400 nm, which was negatively correlated to DOC molecular weight[55]. Fluorescence excitation-emission matrices (EEMs) were measured with an F-4500 fluorometer (Hitachi Ltd., Tokyo, Japan). To minimize inner filter effects, samples were diluted with Milli-Q water until the absorbance at 254 nm was less than 0.3[56]. Based on the EEMs' data, BIX and humification index (HIX) were calculated. Specifically, BIX, an indicator of autotrophic productivity, was expressed as the ratio of emission intensity at 380 nm divided by 430 nm at excitation 310 nm[39]. HIX, representing the extent of DOC degradation, was calculated by dividing the total intensity of emission fluorescence in the range of 435–480 nm by the total intensity of emission fluorescence in the range of 300–445 nm under the excitation wavelength of 254 nm[56].

## FT-ICR MS analysis

To explore the effects of light and microbial degradation on the molecular composition of DOC in thermokarst lakes, we determined the molecular composition of DOC after the light experiment and after 28 days of water incubation using FT-ICR MS, respectively. Specifically, the water samples after light and microbial degradation were first acidified with hydrochloric acid to pH 2. PPL solid-phase extraction (SPE) was then used to remove impurities for FT-ICR MS analysis[57]. The extraction efficiency on the PPL column was 62.29 ± 11.99% (Supplementary Table 4). Then, the sample was continuously injected via a syringe pump at a flow rate of 120 μL/h into a 15.0 Tesla Bruker Solarix FT-ICR MS system equipped with negative electrospray ionization (ESI) ion source[58]. The ESI needle voltage was set to −4.0 kV. 300 single scans with an ion accumulation time of 0.06 s were recorded over a range of mass-to-charge ratio (*m/z*) 100 to 1000. After internal calibration, the detected mass errors are all less than 1 ppm. Peaks were determined using Bruker data analysis software (Bruker Compass DataAnalysis 4.2). According to criteria with elemental combinations of $C_{0-\infty}H_{0-\infty}O_{0-\infty}N_{0-1}S_{0-1}$, all possible molecular formulas were calculated by the molecular formula calculator (Bruker Compass DataAnalysis 4.2) and assigned to the signal-to-noise ratio (S/N) > 4 peaks. In addition, 72.64 ± 2.41% of the peaks were common to the replicates of the same sample (Supplementary Table 4). In the mass spectral peaks with *m/z* values in the range of 100–800 Da, the molecular formula calculation was further restricted according to the principle of 0.3 < H/C < 2.2, O/C < 1.2[59], and the least number of non-oxygen heteroatoms (if an *m/z* value corresponds to more than one possible molecular formula, the molecular formula containing the least number of heteroatoms (N + S) was retained)[60,61].

For FT-ICR MS data, double bond equivalence (DBE)[59], the nominal oxidation state of carbon (NOSC)[62], and the modified aromaticity index ($AI_{mod}$)[63] of each compound were calculated as follows:

$$DBE = 1 + \frac{1}{2} \times (2C - H + N) \tag{1}$$

$$NOSC = 4 - \left( \frac{4C + H - 3N - 2O - 2S}{C} \right) \tag{2}$$

$$AI_{mod} = \frac{DBE_{AI}}{C_{AI}} = \frac{1 + C - \frac{1}{2}O - S - \frac{1}{2}(N + H)}{C - \frac{1}{2}O - N - S} \tag{3}$$

where C, H, O, N and S refer to the number of atoms per formula of carbon, hydrogen, oxygen, nitrogen, and sulfur, respectively. Meanwhile, four classes of compounds were identified based on the criteria of $AI_{mod}$ and H/C[32]: combustion-derived polycyclic aromatics (CA, $AI_{mod} > 0.66$), vascular plant-derived polyphenols (Pol., $0.66 \geq AI_{mod} > 0.50$), highly unsaturated and phenolic compounds (Uns., $AI_{mod} \leq 0.50$, H/C < 1.5) and aliphatic compounds (Ali., $2.0 \geq H/C \geq 1.5$).

## Microbial community structure and function analysis

To evaluate the overall microbial properties across the thermokarst lakes, we determined the community structure, diversity and potential functions of prokaryotic and fungal communities based on the 16S and ITS rRNA gene sequencing. Microbial DNA was extracted using the PowerWater DNA Isolation Kit (MoBio Laboratories, Carlsbad, USA) after filtering ~1000 mL water sample through a 0.22 μm filter (Sterivex GP 0.22, Millipore). DNA concentration and quality were determined with a NanoDrop-8000 UV-Vis Spectrophotometer (Thermo Fisher Scientific, Madison, USA). To explore the prokaryotic and fungal communities, the V4 region of the 16S rRNA gene and the ITS rRNA gene were amplified by the primers 515 F (5′-GTGCCAGCMGCCGCGGTAA-3′), 806 R (5′-GGACTACHVGGGTWT

CTAAT-3′), and ITS3-F (5′-GCATCGATGAAGAACGCAGC-3′) and ITS4-R (5′-TCCTCCGCTTATTGATATGC-3′), respectively[64,65]. Afterwards, polymerase chain reaction (PCR) amplification of 16S rRNA genes was performed in the system of 3 μL DNA (20 ng μL⁻¹), 25 μL 2 × Premix Taq DNA polymerase (Takara Biotechnology, Dalian, China), 1 μL of each primer (10 μM) and 20 μL of nuclease-free water in the instrument of BioRad S1000 (Bio-Rad Laboratory, USA). The thermal cycle conditions were as follows: 5 min at 94 °C for initialization, 30 cycles of 30 s denaturation at 94 °C, 30 s annealing at 52 °C, and 30 s extension at 72 °C, followed by a final elongation at 72 °C for 10 min. PCR amplification of the fungal ITS rRNA gene was conducted using the same system as the 16S rRNA gene, under the thermocycling procedures: 94 °C for 3 min, 35 cycles of 94 °C for 10 s, 52 °C for 10 s, 72 °C for 45 s, and a final extension at 72 °C for 5 min. The concentration and the length of PCR products were identified using a 1% agarose gel electrophoresis. Triplicate PCR products were mixed and purified with ENZA Gel Extraction Kit (Omega Bio-Tek Inc., Norcross, GA, USA), and the sequencing library was generated using NEBNext ® Ultra™ DNA Library Prep Kit for Illumina (New England Biolabs, Ipswich, MA, USA) following manufacturer's recommendations. The library quality was assessed by Qubit 3.0 Fluorometer (Life Technologies, Grand Island, 165 NY) and Agilent 4200 (Agilent, Santa Clara, CA) system. At last, the products were sequenced using an Illumina Nova 6000 platform (Guangdong Magigene Biotechnology Co., Ltd) to obtain 2 × 250 bp paired-end reads.

The data were analyzed on the Magigene Cloud Platform (http://cloud.magigene.com). Sliding window (window size = 4, quality = 20) in the fastp software (version 0.14.1, https://github.com/OpenGene/fastp) was used to check the quality of the acquired raw data[66]. The primers were removed from the paired-end clean reads using cutadapt software (V1.14, https://github.com/marcelm/cutadapt/). Then, the obtained paired-end clean reads were merged by usearch (V10, http://www.drive5.com/usearch/) based on the overlap between the paired-end reads. All quality-filtered sequences from 16S rRNA and ITS rRNA gene amplicons were clustered into Operational Taxonomic Units (OTUs) at 97% similarity cutoff using UPARSE (version 7.1, http://drive5.com/uparse)[67]. For each OTU, the most commonly occurring sequence was selected as the representative sequence and screened for further annotation. The taxonomy was assigned to each representative sequence using the SILVA v132 (16S rRNA, http://www.arb-silva.de) and Unite v8.0 (ITS rRNA, http://unite.ut.ee) database at a confidence cutoff of 80%[68,69]. The Richness index was calculated to evaluate the alpha diversity of the microbial community. Furthermore, FAPROTAX[35] and FUNGuild[36] were used to predict prokaryotic and fungal functions based on OTU taxonomic information, respectively.

## Statistical analysis

Statistical analyses mainly consisted of the following steps after normalizing the data by log transformation when necessary. First, two-way ANOVAs were used to explore whether and how light exposure and lake location affected water physical properties and DOC chemistry in the UV experiment, with UV treatment (UV exposure vs. control) and lake location as fixed effects. Meanwhile, a three-way ANOVA was performed to assess the effects of UV treatment, incubation temperature, lake location and their interactions on microbial $CO_2$ production. Given the response of microbial respiration to UV exposure was independent of the incubation temperature, indicated by the insignificant interaction between UV treatment and incubation temperature (Supplementary Table 1, $P = 0.49$), we pooled the data on microbial $CO_2$ production from two temperatures together in the subsequent analysis.

Second, to explore the driving factors of $CO_2$-RR (i.e., microbial respiration response to UV light) across the thermokarst lakes,

regression analyses were conducted to clarify the associations of $CO_2$-RR with DOC physical properties, DOC chemistry, and its molecular composition. Of them, the first component (PC1) of principal components analysis (PCA) for DOC and DON content was used to represent DOM content, and its relationship with DOC chemistry was also examined by regression analyses. Additionally, given the significant changes in DOC chemistry after UV light exposure, the associations of $CO_2$-RR with DOC chemistry-RR, photo-produced and photo-degraded compound properties were further explored to reveal the mechanisms underlying the sunlight effect on microbial respiration. Specifically, for FT-ICR MS data, the change in peak intensity was considered significant if the absolute intensity and the relative intensity were > 20% after UV light exposure or microbial degradation[25]. Based on this rule, formulas were categorized as photo-produced, photo-degraded and biodegraded compounds. Meanwhile, to verify the rationality of this criterion, we also categorized the photo-produced, photo-degraded, and bio-degraded compounds as the formulae only present or absent following photochemical or microbial degradation, respectively. The strong correlations of the properties of photo-produced, photo-degraded, and bio-degraded compounds between the two methods demonstrated the rationality of the criteria we employed (Supplementary Fig. 13).

Finally, to elucidate the potential functions of microbes in thermokarst lakes, we used Wilcoxon rank-sum test to compare the relative abundance of different functional groups in the prokaryotic and fungal communities. Meanwhile, to confirm these microbial functional predictions, we compared the chemical properties of bio-degraded compounds with photo-produced and photo-degraded compounds. The proportion of four types of DOC compounds (i.e., Ali., Uns., Pol., CA) that were degraded by microbes during the incubation was also compared using one-way ANOVA. Statistical differences were considered to be significant at the level of $P < 0.05$. All the statistical analyses were performed in R (version 4.0.5, R Core Team, 2021).

### Reporting summary

Further information on research design is available in the Nature Portfolio Reporting Summary linked to this article.

## Data availability

All prokaryotic and fungal sequences have been deposited in NCBI's SRA database under project accession numbers PRJNA948140 and PRJNA948167. All data required to reproduce the results are available in the Figshare database (https://doi.org/10.6084/m9.figshare.22331278)[70].

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

## Acknowledgements

We are grateful to Dr. Dong Cao and Dr. Jitao Lv at Research Center for Eco-Environmental Sciences, Chinese Academy of Sciences for Fourier transform ion cyclotron resonance mass spectrometry analysis. This work was supported by the National Key R&D Program of China (2022YFF0801903), CAS Project for Young Scientists in Basic Research (YSBR-037), National Natural Science Foundation of China (32241034 and 42141006), the Second Tibetan Plateau Scientific Expedition and Research (STEP) program (2019QZKK0302), Youth Innovation Promotion Association of the Chinese Academy of Sciences (Y2021031).

## Author contributions

L.C. designed the research. J.H., X.F., and Z.W. performed the research. L.K., Z.L., W.Z., and X.L. performed the field sampling. L.C., J.H. and C.L. analyzed the data. L.C., J.H., and Y.Y. wrote the paper with input from other authors.

## Competing interests

The authors declare no competing interests.
