## [Peer Review File · Nature Communications]

Photo-produced aromatic compounds stimulate microbial degradation of dissolved organic carbon in thermokarst lakesReviewer #1 (Remarks to the Author):

Review of Photo-produced small aromatic compounds stimulate microbial degradation of dissolved organic carbon in thermokarst lakes

Summary

This manuscript investigates the effect of sunlight exposure on the microbial degradation of dissolved organic carbon (DOC) in thermokarst lakes. The "photo-bio" degradation of DOC has long been known to be an important control on DOC biogeochemistry (e.g., Wetzel 1995) and on CO₂ emissions from arctic, boreal and temperate freshwaters (as shown by papers cited in this manuscript). The objective of this manuscript is to test ideas shown in the past ~ 5-10 years on why sunlight generally causes microbes to respire more DOC to CO₂ compared to the same DOC kept in the dark. The manuscript builds nicely on the ideas in the literature, often showing many of the same results as the papers cited in the manuscript. The manuscript appears to contain high quality and carefully collected data.

However, there are many unclear or missing methodological details that make it difficult for a reader to assess the approach and thus assess how rigorous are the results and conclusions. In addition, in some cases the interpretation of the results is too simplistic or superficial (see detailed comments below). Thus, the manuscript could be substantially improved if the authors took more caution and care to interpret their data by clearly listing assumptions and more clearly describing what was (or wasn't) measured.

In addition, it seems that the authors may not have the data to make a quantitative case for the importance of photo-bio degradation of DOC as a source of CO₂ from these lakes (e.g., from a mass balance or rate comparison argument). That limitation is ok, considering the strength of this manuscript is in the qualitative (by its nature) chemical characterization of the impacts of sunlight on DOC composition and in turn, CO₂ respired. The authors should be clear about this limitation up front in the introduction, so as not to leave a reader hanging in the conclusions wondering how important this process is in these lakes (which has been quantified in other regions, see detailed comments below). My suggestion to strengthen the manuscript is to set up the knowledge gap in the introduction as not knowing whether the "new view" of the effect of sunlight on microbial respiration of DOC would apply in thermokarst lakes (which IS the specific knowledge gap addressed in this study, where by new view I mean the conclusions of Ward et al. 2017, Nalven et al. 2020 among other papers cited in the manuscript). The authors could discuss why this new view would or would not be expected to apply in thermokarst lakes of the Tibetan Plateau. Set up in this manner would allow the authors to remove the quantitative justification from the introduction. Alternatively, if the authors have the data to make mass balance or rate comparisons they should add that data and interpretation to the manuscript.

Detailed comments

Line 47: "most active C pool" is highly subjective to a specific biogeochemical process. Please revise to be more specific about the role of DOC as a source of CO₂ from these lakes.

Line 62: Replace molecule with compound which is a more precise term than molecule in this context given that what is known is only amount of CHNOS in DOM not how these atoms are arranged into structures.

Line 72-77: Please revise these statement to accurately reflect the extensive literature on the biological and photochemical degradation of permafrost DOC (e.g., as cited in this manuscript Cory et al. 2013 PNAS, Bowen et al. 2020 GRL along with many other papers in the literature).

Line 82-85: Here the authors need to distinguish between light availability and rates of light absorption in the water column. The authors here imply that DOC photochemistry may be important in thermokarst lakes of the Tibetan Plateau based only on the surface photon flux. However, there could be a high photon flux but if the concentration of light-absorbing dissolved

organic matter (i.e., CDOM) in the lake is too low in the water to absorb the sunlight, rates of any photochemical process would be too low to be important sources of CO₂ from the lake. This is because the rate of any photochemical process in the environment (such as a lake) depends on both the light available and the concentration of light-absorbing species (see Cory and Kling 2018, *Limnology & Oceanography Letters* for a review). The authors appear to have CDOM concentrations for the lakes in Supplementary Dataset 1. Thus, their manuscript would be substantially strengthened by comparing rates of light absorption by CDOM in their lakes (i.e., surface photon flux spectra x CDOM spectra) to rates of light absorption by CDOM in other freshwaters where DOC photochemistry has already been shown to be an important source of CO₂ from lakes (e.g., Vahatalo et al. 2000 *Limnology and Oceanography* 2000, Cory et al. 2014 *Science*, Bowen et al. 2020 in *Limnology and Oceanography* for arctic, boreal and temperate waters). Without knowing the CDOM data (and rates of light absorption in the water column), it is impossible for a reader to assess whether CDOM in these lakes is “vulnerable” to photodegradation as the authors claim.

Line 111-112: Why were 10 and 20 °C chosen for the incubations? How representative are these temperatures of the thermokarst lakes? Do these temperatures refer to the air or water temperature over the course of the experiment?

Line 139: The authors should take more care and caution here in ruling out the role of microbial community shifts or reactive oxygen species (ROS) on the respiration results. Without data showing no difference in microbial community composition or reactive oxygen species (ROS) between treatments the authors can't rule out the impact of shifts in microbial community composition and ROS on the light vs. dark difference in respiration. The authors cite papers in support of photochemical changes in DOM composition as more likely to impact light vs. dark differences in respiration, but a careful read of those papers shows that the effect of community composition and ROS are critical discussion points and not settled knowledge. In addition, one of the manuscripts cited (25) plus other papers (e.g., Cory et al. 2013 *PNAS*) show statistically significant differences in microbial community composition between the inoculum and between the microbial communities growing on light-exposed vs. dark control DOM.

Line 174: DOM content is not related to the chemistry of photo-degraded compounds? What does this statement mean? That there was no correlation between the concentration of DOM (i.e., DOC) and any of the mass spec data of the photo-degraded compounds? Because FTICR mass spec data is not quantitative (e.g., Hockaday et al. 2009 in *Limnology and Oceanography*, Sleighter et al. 2012 in *Analytical Chemistry*, among many other papers), one would not expect a correlation between concentration (on a mass basis such as DOC) and mass spec signals.

Line 191-193: The author's data do contrast the traditional view, as do the papers cited in this section of the manuscript. Thus, the authors could more accurately describe their contribution as building on the newer view of sunlight's effects on DOM photodegradation by reproducing the same observations as others (e.g., Ward et al. 2017 in *Nature Communications*, Nalven et al. 2020 in *Env. Micro.*, Bowen et al. 2019 in *Limnology & Oceanography*) but in a different region (e.g., thermokarst lakes vs. arctic or temperate freshwaters in these other studies).

Line 210: “the enrichment of these taxa” is confusing, in what lake or treatment where these taxa enriched? What is the baseline for comparison here?

Line 213-225 and Figure 3: This section and figure is confusing. What is being compared here? The relative abundance of microbes that break down aromatics vs cellulose? Is the comparison across the 10 thermokarst lakes, or between light and dark treatments? It's not clear that there is a light vs. dark control difference here in microbial community composition or function, so it is not clear how these data support (or do not support) the effect of sunlight on microbial respiration of DOC.

Line 256: It is not accurate to say that this manuscript uncovered a mechanism of response of DOC biodegradation in response to sunlight given that the paragraph above (lines 239-253) reports nearly identical results to published literature (e.g., Ward et al. 2017 in *Nature Communications*). A more accurate summary would be to say that this manuscript here builds on prior work by showing that this newer view of photo and biodegradation of DOC appears to explain

DOC degradation in thermokarst lakes on the Tibetan Plateau.

Line 268: Please revise "more complicated than originally anticipated". There is no paper suggesting that DOC degradation is "not complicated" (and it is not possible to define "complicated").

Lines 256-279: Can the authors do more to put their results into context of regional carbon budgets? For example, how significant is the potential for photo-bio degradation of DOM to the CO₂ released from these lakes? Given that the lakes are generally located in swamp meadows, is there any shading of lake water by vegetation that would limit the importance of photo-bio degradation? The authors start out with a quantitative argument in the introduction (lines 39-57), thus the manuscript reads as incomplete by not returning to the quantitative justification for the work in the conclusions and implications section.

Conceptual figure 5: What is the cause of the "increase of DOM content and terrestrially-derived DOC"? Please note inconsistent use of DOC and DOM, molecule and compound throughout the manuscript. It is also not clear what is the difference between the left and right panel in this figure? I think the left panel is indicating less CDOM (i.e., less aromatic DOM), and thus less UV enhancement of DOM biodegradation compared to the right panel? How many lakes in the region fall into these two categories?

Methods text:

Overall, it is not clear what the CO₂-RR (response ratio) is? Line 100-101 indicates it is a ratio between the sunlight treatment and the "control group" but the methods section does not clarify what is the control group?

Line 295: Was the GF/C filtered inoculum frozen at -20 °C prior to the incubation? If so, what effect did that storage method have on the microbial communities?

Line 307: What was the time between sample collection and filtration and the start of the incubation experiment? I.e., how long were samples stored at -20°C? Were the samples thawed prior to the start of the incubation and how long did that take? Was it done in the dark? What was the temperature during the UV light exposure treatment and did this temperature differ from the dark control water?

Line 314: At what wavelength(s) is the spectral intensity compared between the UVA bulbs and the solar irradiance reaching the Tibetan Plateau? 340 nm? Is the 9 hour comparison here saying that 21 hours of experimental light at 340 nm would take 9 hours for the same 340 nm light to reach the surface of the lakes? If so, this is a helpful but incomplete comparison. A more rigorous and helpful comparison would be to compare the rate of light absorbed by the CDOM in the lake vs. the experiment (see Vahatalo et al. 2000 in *Limnology and Oceanography*, Vahatalo and Wetzel 2008 in *Limnology and Oceanography*, Cory et al. 2014 in *Science*). Given that the authors have the CDOM data as reported in Supplementary Dataset 1, they can make these calculations.

Line 325: ROS quenching was assumed to take place over 12 hours but ROS were not measured, correct? Please clarify what was measured vs. assumed.

Line 326: Why was the inoculum pre-incubated at 20 °C for 7-14 days prior to the experimental incubation? Can the authors discuss the potential bottle effects that change microbial communities and their activities over this time?

Line 327: What was the purpose of the sterile water blanks? Were these blanks analyzed by FTICR or for respiration or some other purpose?

Line 325: What was the "respective inoculum" in each sample? Was it the GF/C filtered water from the lake corresponding to the light-exposed and dark GF/F filtered water?

Line 329: Is there data to support that the flushing with CO₂ free air result in similar initial CO₂ concentrations in the water at the start of the incubation?

Line 340-344: How is the experimental incubation different here when measuring composition of biodegraded compounds vs. the incubations to measure respiration (CO₂ in the headspace) described above? In addition, what is the control for the analysis of biodegraded DOC? That is, when doing FTICR analysis of DOM, it is very challenging to determine an effect of a treatment given the qualitative nature of this analysis combined with less than 100% extraction efficiency by PPL. In addition, there is biodegradation of DOC occurring in the inoculum which may dilute any treatment signal (e.g., the effect of light in this case). While the authors indicated no significant loss of DOC during the light-exposure portion of the experiment, differences in extraction efficiency between samples would be expected given the large changes in DOM composition, and these differences in extraction efficiency may account for a lot of difference in sample spectra analyzed by FTICR mass spec.

Line 376: What was the extraction efficiency on the PPL column? What was the reproducibility of the FTICR spectra? For example, were samples analyzed in duplicate or triplicate and if so, how many masses were common to the replicates? (See Sleighter et al. 2012 in *Analytical Chemistry*, Ward and Cory 2015 in *GCA*).

Reviewer #2 (Remarks to the Author):

This paper presents a study of the combined controls of sunlight exposure and microbial degradation on dissolved organic carbon mineralization in lakes formed by thermokarst failures of permafrost. These lakes vary in DOC content and serve as important conduits for the flux of carbon from land to atmosphere. The authors conducted incubation experiments in which they photo-exposed filter-sterilized DOC from a range of lakes with the equivalent of ~ one day of sunlight, and then inoculated that organic matter with bacteria from its lake of origin. They found many remarkably strong relationships between DOC chemistry and the effect of sunlight exposure on the degree to which sunlight enhanced or reduced microbial respiration. They concluded that photo-production of small aromatic compounds was the main mechanism by which sunlight enhanced microbial respiration. They also concluded that aromatic-compound degrading microbial taxa were abundant in their samples.

I think this was a well-conceived, high-quality study that found very strong relationships that are potentially ground-breaking. The methodology is sound, and the methods provide adequate detail. This work will be significant for the field.

My main concerns are that:

1. The Results/Discussion section is difficult to understand for a non-expert in aquatic organic carbon chemistry. This is in part unavoidable because of the nature of this complicated experiment. But it is also because of the huge number of abbreviations, most of which readers are expected to understand and remember.
2. Conclusions concerning patterns in microbial community composition and predicted functions appear to be overstated and potentially misleading. First, the authors analyzed microbial diversity of the inocula used for the experiments and not the experimental incubations, but in several places the text uses the word "enriched" making it sound like the results are from the experimental incubations. Second, many of these taxa are very common in freshwater ecosystems, but for some reason the authors describe them as "enriched" in thermokarst lakes. Third, some of the text implies that there is a relationship between potential aromatic-degrading taxa and DOM chemistry measurements. For example, in Fig. 5 they show a gradient bar labeled "Microbes that degraded aromatic compounds" right next to a similar gradient bar labeled "Light enhancement on DOC biodegradation." I cannot find where the authors show this relationship in their results. Fourth, the abstract says DOC degradation is "attributed to the high relative abundance and diversity of bacteria and fungal taxa that have great abilities to decompose complex compounds," but these taxa are abundant in many different freshwater systems, and so this attribution seems suspect.

Abstract

Lines 25-27. I see where there are positive relationships between CO₂-RR and the abundance of

photoproducted aromatic compounds, but I don't see where there are negative relationships between CO₂-RR and the abundance of photoproducted aliphatic compounds. This is not clearly explained and defended in the text.

Lines 28-30. These appear to be typical freshwater taxa, and it is not clear why the authors would describe their abundances as "relatively high."

Line 31. I thought FT-ICR MS only effectively captures larger molecular weight compounds, and that small compounds such as sugars and amino acids are not easily detectable. I think it is important that the authors define what they mean by "small" compounds here.

Introduction.

Line 39. I don't understand the odd superscript and subscript for the number 10¹⁴.

Results and discussion

Lines 137-140. I don't understand this sentence.

Lines 141-144. Supplementary figures 3a-d shows some significant changes in DIC, DON, and pH in some lakes following sunlight exposure. It seems incorrect to say that sunlight did not affect these measurements.

Lines 162-167. This seems like an important point, but it is very difficult to understand, in part because the sentence includes 11 abbreviations.

Lines 176-178. The word "specifically" suggested this sentence was going to describe details of the relationships described in the previous sentence about Fig. 2a (chemistry measurements vs. "DOM chemistry-RR"), but instead it is about something else. I don't understand what the authors are implying here.

Line 184. This sentence implies that DBE (double bond equivalence) and NOSC (nominal oxidation state of carbon) are measurements of small aromatics, but I don't think they are.

Lines 199-200. I don't understand this sentence. It seems to assume that smaller and less aromatic DOC compounds are always less abundant than larger and more aromatic DOC compounds. Is that true?

Lines 204-209. Burkholderiales are dominant members of most freshwater bacterioplankton communities, and the other taxa listed here are also common bacterioplankton. I don't understand how these taxa could be described as "enriched" in thermokarst lakes. What are the authors comparing the abundances to?

Lines 229-231. I don't understand this sentence.

Lines 233-235. I don't understand what the authors mean here or why it is relevant.

Lines 261-264. I think the gradient line on Fig. 5 labeled "Microbes that degraded aromatic compounds" is misleading. The authors did not show a relationship between the abundance of these microbes and the degree of light enhancement on DOC biodegradation.

Methods

Line 326. Why were microbes "pre-incubated"? Is it more appropriate to say they were "stored" for 7 days while the DOC was processed? If there was a scientific justification for this "pre-incubation" then it should be presented here.

Line 410. Why were the microbes pre-incubated for 14 days rather than 7 days like the inocula? Is there a scientific justification for this?

End of Review

Responses to Reviewer #1

[Comment 1] This manuscript investigates the effect of sunlight exposure on the microbial degradation of dissolved organic carbon (DOC) in thermokarst lakes. The “photo-bio” degradation of DOC has long been known to be an important control on DOC biogeochemistry (e.g., Wetzel 1995) and on CO₂ emissions from arctic, boreal and temperate freshwaters (as shown by papers cited in this manuscript). The objective of this manuscript is to test ideas shown in the past ~ 5-10 years on why sunlight generally causes microbes to respire more DOC to CO₂ compared to the same DOC kept in the dark. The manuscript builds nicely on the ideas in the literature, often showing many of the same results as the papers cited in the manuscript. The manuscript appears to contain high quality and carefully collected data.

[Response] Many thanks for the reviewer’s excellent comments. These comments listed below help us to conduct a thorough revision of the manuscript. We really appreciate this professional review which greatly improved our paper.

Major comments:

[Comment 2] However, there are many unclear or missing methodological details that make it difficult for a reader to assess the approach and thus assess how rigorous are the results and conclusions. In addition, in some cases the interpretation of the results is too simplistic or superficial (see detailed comments below). Thus, the manuscript could be substantially improved if the authors took more caution and care to interpret their data by clearly listing assumptions and more clearly describing what was (or wasn’t) measured.

[Response] Thanks for the reviewer’s insightful and enlightening comments. These constructive comments, together with those listed below have guided us to conduct a thorough revision of the original manuscript. Following the reviewer’s comments, we have clarified and added more methodological details (e.g., the basis for the selection of incubation temperature (Comment 8), the duration and temperature of sample storage and processing (Comments 19, 20, 23), the extraction efficiency on the PPL column

(Comment 28), the reproducibility of the FT-ICR MS (Comment 28), and so on) to facilitate the reader's evaluation in the revised MS (Page 17, line 354-361; Page 15, line 319-323; Pages 16-17, line 341-350; Page 20, line 413-414; Page 20, line 424-425). Additionally, to further elaborate on our results, we have clearly described that we did not measure ROS content during the incubation (Comment 22). Instead, ROS quenching was assumed to take place over 12 hours of placement before the biodegradation experiment in the revised MS (Page 7, line 138-141; Page 16, line 337-340). For detailed modifications please see our responses to the following comments.

[Comment 3] In addition, it seems that the authors may not have the data to make a quantitative case for the importance of photo-bio degradation of DOC as a source of CO₂ from these lakes (e.g., from a mass balance or rate comparison argument). That limitation is ok, considering the strength of this manuscript is in the qualitative (by its nature) chemical characterization of the impacts of sunlight on DOC composition and in turn, CO₂ respired. The authors should be clear about this limitation up front in the introduction, so as not to leave a reader hanging in the conclusions wondering how important this process is in these lakes (which has been quantified in other regions, see detailed comments below). My suggestion to strengthen the manuscript is to set up the knowledge gap in the introduction as not knowing whether the “new view” of the effect of sunlight on microbial respiration of DOC would apply in thermokarst lakes (which IS the specific knowledge gap addressed in this study, where by new view I mean the conclusions of Ward et al. 2017, Nalven et al. 2020 among other papers cited in the manuscript). The authors could discuss why this new view would or would not be expected to apply in thermokarst lakes of the Tibetan Plateau. Set up in this manner would allow the authors to remove the quantitative justification from the introduction. Alternatively, if the authors have the data to make mass balance or rate comparisons they should add that data and interpretation to the manuscript.

[Response] Very good suggestion! We do agree with the reviewer that the strength of our manuscript is in the qualitative chemical characterization of the sunlight effects on

DOC composition and CO₂ respired rather than in the quantitative analysis. Moreover, we don't have the data required to make a quantitative analysis. Thus, **following the reviewer's suggestions, we have weakened the quantitative statements in the Introduction session to remove the quantitative justification** (Pages 2-3, line 33-48).

Meanwhile, we have set up the knowledge gap as not knowing which of the two alternative views on the effect of sunlight on microbial respiration of DOC would apply in the thermokarst lakes. Furthermore, based on the specificities of the DOC properties in thermokarst lakes, **we have also proposed a hypothesis and discussed why the emerging view would be expected to apply in the thermokarst lakes** as follows: “*Despite all the extensive empirical progress in arctic and temperate ecosystems (Bowen et al., 2020a; Cory et al., 2013; Tranvik et al., 2001; Ward et al., 2017), the evidence for the biological and photochemical degradation of DOC in thermokarst lakes is still lacking. It remains unknown which of the two alternative views would apply in this unique ecosystem. It was reported that microbes in the permafrost DOC trended to degrade more aromatic and oxidized DOC along permafrost thawing gradients on the plateau (Zhou et al., 2020). Meanwhile, the dominance of terrestrial-derived aromatic compounds in thermokarst lakes would increase with permafrost thawing (Cory et al., 2013; Wauthy et al., 2018). Therefore, aromatic compounds rather than low molecular weight and aliphatic-like DOC compounds were expected to stimulate microbial respiration of DOC in thermokarst lakes. Here, we hypothesized that consistent with the emerging view (Bowen et al., 2020b; Nalven et al., 2020; Ward et al., 2017), sunlight may promote microbial degradation by converting high molecular weight aromatic-like DOC into more biodegradable aromatic compounds in thermokarst lakes*” (Page 4, line 65-77).

Detailed comments:

[Comment 4] Line 47: “most active C pool” is highly subjective to a specific biogeochemical process. Please revise to be more specific about the role of DOC as a

source of CO₂ from these lakes.

[Response] Following the reviewer's comments, we have revised the sentence as follows: "*Dissolved organic C (DOC) is an important source of this C release, and its decomposition was jointly controlled by sunlight and microbial processing (Cory et al., 2018; Vonk et al., 2015a)*" (Page 3, line 40-42).

[Comment 5] Line 62: *Replace molecule with compound which is a more precise term than molecule in this context given that what is known is only amount of CHNOS in DOM not how these atoms are arranged into structures.*

[Response] Following the reviewer's comment, we have replaced the word "molecule" with "compound" in the revised manuscript.

[Comment 6] Line 72-77: *Please revise these statement to accurately reflect the extensive literature on the biological and photochemical degradation of permafrost DOC (e.g., as cited in this manuscript Cory et al. 2013 PNAS, Bowen et al. 2020 GRL along with many other papers in the literature).*

[Response] Sorry for the inappropriate statement. Following the reviewer's suggestions, we have rephrased the sentences as follows: "*Despite all the extensive empirical progress in arctic and temperate ecosystems (Bowen et al., 2020a; Cory et al., 2013; Tranvik et al., 2001; Ward et al., 2017), the evidence for the biological and photochemical degradation of DOC in thermokarst lakes is still lacking*" (Page 4, line 65-67).

[Comment 7] Line 82-85: *Here the authors need to distinguish between light availability and rates of light absorption in the water column. The authors here imply that DOC photochemistry may be important in thermokarst lakes of the Tibetan Plateau based only on the surface photon flux. However, there could be a high photon flux but if the concentration of light-absorbing dissolved organic matter (i.e., CDOM) in the lake is too low in the water to absorb the sunlight, rates of any photochemical process*

would be too low to be important sources of CO₂ from the lake. This is because the rate of any photochemical process in the environment (such as a lake) depends on the both the light available and the concentration of light-absorbing species (see Cory and Kling 2018, *Limnology & Oceanography Letters* for a review). The authors appear to have CDOM concentrations for the lakes in Supplementary Dataset 1. Thus, their manuscript would be substantially strengthened by comparing rates of light absorption by CDOM in their lakes (i.e., surface photon flux spectra x CDOM spectra) to rates of light absorption by CDOM in other freshwaters where DOC photochemistry has already been shown to be an important source of CO₂ from lakes (e.g., Vahatalo et al. 2000 *Limnology and Oceanography* 2000, Cory et al. 2014 *Science*, Bowen et al. 2020 in *Limnology and Oceanography* for arctic, boreal and temperate waters). Without knowing the CDOM data (and rates of light absorption in the water column), it is impossible for a reader to assess whether CDOM in these lakes is “vulnerable” to photodegradation as the authors claim.

[Response] Very good suggestion! We are grateful to the reviewer for providing this valuable comment. By carefully reading these references provided by the reviewers (Bowen et al., 2020b; Cory et al., 2014; Cory et al., 2018; Vahatalo et al., 2000), we have tried to calculate the rates of light absorption by CDOM ($Q_{a,\lambda}$) in the 10 thermokarst lakes (Fig. R1) as follows (Bowen et al., 2020b):

$$Q_{a,\lambda} = \int_{\lambda_{\min}}^{\lambda_{\max}} E_{\lambda} \left(1 - e^{-a_{CDOM,\lambda} \times z}\right) \frac{a_{CDOM,\lambda}}{a_{tot,\lambda}} d\lambda \quad (1)$$

where λ_{\min} and λ_{\max} are the minimum and maximum wavelengths of UV light absorbed by CDOM (280 nm and 400 nm, respectively). E_{λ} is the photon flux spectrum (mol photon m⁻² nm⁻¹). The fraction of light absorbed by CDOM relative to other light-absorbing constituents, $a_{CDOM,\lambda}/a_{tot,\lambda}$, was often equal to 1 at all wavelengths (Cory et al., 2014). We assumed that the path length of light (z) was equivalent to the average depth of the lakes (ranging from 0.24 m to 0.68 m) in the field. The results showed that the rates of light absorption by CDOM ($Q_{a,\lambda}$) ranged from 95.10 to 153.21 $\mu\text{mol photon m}^{-2} \text{s}^{-1}$ in the 280-400 nm across the 10 lakes (Fig. R1).

Nevertheless, it should be noted that we only have the data of total surface solar radiation intensity (200-4000 nm, W m^{-2}) (Tang, 2019), but no data of photon flux spectrum (E_λ , $\text{mol photon m}^{-2} \text{ nm}^{-1}$) for the Tibetan Plateau. Thus, **the E_λ of the 10 thermokarst lakes was predicted by the NREL SMARTS model** (Simple Model of the Atmospheric Radiative Transfer of Sunshine, V.2.9.5) (Cory et al., 2014). Specifically, to obtain the solar spectrum of these lakes, several atmospheric inputs, *e.g.*, atmospheric pressure, relative humidity, ozone abundance, and aerosol optical depth, are required in the model. Since there were no meteorological stations near these 10 thermokarst lakes, we had to choose the reanalysis data from satellites as the input data for the model (Table R1). However, **due to the strong elevation effects on aerosols and water vapor** (Gueymard et al., 2018; Gueymard, 2019), **these reanalysis data would lead to large uncertainties in estimating E_λ and $Q_{a,\lambda}$ across the Tibetan Plateau**. Apart from this uncertainty induced by the input data, **divergent spectral ranges used in different studies** (*e.g.*, 280-400 nm in our thermokarst lakes, 280-600 nm, 280-700 nm, 300-700 nm in other freshwaters) (Bowen et al., 2020b; Cory et al., 2014; Vahatalo et al., 2000) **could further lead to uncertainties in comparing the rates of light absorption by CDOM among different lakes**. Due to these two points, we preferred not to put these results in the main text. To avoid confusion, we have removed the relevant argument about the vulnerability of DOC to photodegradation on the Tibetan Plateau in the revised MS. Thanks for your understanding!

Fig. R1. Sunlight absorbed by CDOM in 10 thermokarst lakes. Different colors represent different lakes. The data in parentheses represent the integral value of sunlight absorbed by CDOM from 280 to 400 nm for each lake ($\mu\text{mol photons m}^{-2} \text{s}^{-1}$). The ten sampling sites are Yushu (YS), Golmud (GM), Nagqu (NQ), Amdo County (AD), Nyainrong County (NR), Seni District (SN), Heihe River (HH), Golog Tibetan Autonomous Prefecture (GL), Madoi County (MD), and Maqên County (MQ).

Table R1. List of reanalysis data from satellites as SMARTS model input data and their sources.

Variable	Publication	Data sources
SPR (mb)	Yang et al. (2019)	https://data.tpdc.ac.cn/zh-hans/data/8028b944-daaa-4511-8769-965612652c49/
RH (%)	Wang (2022)	https://www.tpdc.ac.cn/zh-hans/data/99dd84e2-288d-4db8-b098-8118b3b0c17a
TDAY (°C)	National Earth System Science Data Center, National Science & Technology Infrastructure of China	http://www.geodata.cn/datapplication/OrderStepList.html?dataguid=250085273409240
IH2O (cm)	Dee et al. (2011)	https://doi.org/10.1002/qj.828
TAU550	National Earth System Science Data Center, National Science & Technology Infrastructure of China	http://www.geodata.cn/data/datadetails.html?dataguid=1776940&docId=3316

Notes: SPR, surface pressure; RH, relative humidity; TDAY, the average daily temperature at the site level; IH2O (cm), precipitable water; TAU550, aerosol optical depth at 550 nm, τ_{550} .

[Comment 8] Line 111-112: Why were 10 and 20 °C chosen for the incubations? How representative are these temperatures of the thermokarst lakes? Do these temperatures refer to the air or water temperature over the course of the experiment?

[Response] Very good comment! 10°C was chosen because the daytime mean lake surface water temperature (LSWT) during the growing seasons (*i.e.*, from May to October) in 2001-2015 was estimated at approximately 12°C across the Tibetan Plateau (Wan et al., 2017). Meanwhile, room temperature at 20°C was recommended to perform DOC incubation in permafrost soils and aquatic systems, as this is the most common, relatively easy to maintain, and allows comparison among studies (Vonk et al., 2015b). Therefore, these two temperatures were chosen to investigate the potential temperature effects on DOC biodegradation in our study. During our incubation, these temperatures refer to the air temperature throughout the experiment in the incubator. We have added the additional information in the *Methods* session as follows: “*Then, all bottles were incubated for 28 days in a dark environment (Vonk et al., 2015b) at 10°C and 20°C. To be specific, 10°C was chosen because the daytime mean lake surface water temperature (LSWT) during the growing seasons (i.e., from May to October) in 2001-2015 was estimated at approximately 12°C across the Tibetan Plateau (Wan et al., 2017). Meanwhile, room temperature at 20°C was recommended to perform DOC incubation in permafrost soils and aquatic systems, as this is the most common, relatively easy to maintain, and allows comparison among studies (Vonk et al., 2015b)*” (Page 17, line 353-360).

[Comment 9] Line 139: The authors should take more care and caution here in ruling out the role of microbial community shifts or reactive oxygen species (ROS) on the respiration results. Without data showing no difference in microbial community composition or reactive oxygen species (ROS) between treatments the authors can't rule out the impact of shifts in microbial community composition and ROS on the light vs. dark difference in respiration. The authors cite papers in support of photochemical changes in DOM composition as more likely to impact light vs. dark differences in

respiration, but a careful read of those papers shows that the effect of community composition and ROS are critical discussion points and not settled knowledge. In addition, one of the manuscripts cited (25) plus other papers (e.g., Cory et al. 2013 PNAS) show statistically significant differences in microbial community composition between the inoculum and between the microbial communities growing on light-exposed vs. dark control DOM.

[Response] Very good comment! Sorry for the unclear expressions and the inappropriate citation in the original manuscript. Actually, **to rule out the role of microbes during the UV degradation, we isolated the water samples with sterile 0.22 µm filters (Sterivex GP 0.22, Millipore) before light exposure.** This pore size of the filters was widely used in previous studies assuming that it can eliminate most of the microbial populations (e.g., prokaryotes and fungi) in the water (Abbott et al., 2014; Cory et al., 2013). To test this assumption, it's ideal to compare the microbial cell counts before and after passing through 0.22 µm filters. Nevertheless, given that all the initial water samples (except the microbial inocula) were filtered through Whatman GF/F filters, we could only compare the microbial cell counts after passing through different-size filters (GF/F vs. 0.22 µm filters). **Our results showed that even compared to the GF/F filters, the 0.22 µm filters reduced 87.7% of the microbial populations (measured as microbial cell counts) to the level of Milli-Q water (Table R2).** Despite a slight increase in microbial cell counts observed in dark control after 21h of placement, the difference between light exposure and dark control was not significant after the sunlight experiment (Table R2, $P = 0.18$). Given that the initial microbes were further inoculated before the biodegradation experiment, such slight differences in microbial cell counts after the sunlight experiment could be assumed to be neglected. Regarding the role of ROS, **after the sunlight exposure experiment, the water samples were placed for 12 hours before being used for subsequent biodegradation experiments.** **Based on the previous studies (Ward et al., 2017), the ROS could largely decay during this 12-hour placement.** Thus, we assumed that the influence of ROS on microbes was also minimal.

We have clearly mentioned these points and added the related data in the *Results and Discussions* session of the revised MS as follows: “*Firstly, given that most microbes were eliminated by filtering the water samples through 0.22 μm filters before the sunlight experiment (Supplementary Table 2), sunlight exposure would barely affect the microbes. Additionally, we assumed that the reactive oxygen species (ROS) could largely decay during the 12-h placement after the sunlight experiment (Ward et al., 2017), and its impact on biodegradation was also minimal in our study (See Methods for details)*” (Page 7, line 135-141). We also added detailed information in the *Methods* sessions as follows: “*To remove microbes during the UV degradation, we isolated the GF/F filtered water samples further with 0.22 μm filters (Sterivex GP 0.22, Millipore) to generate triplicate samples for each thermokarst lake. This pore size of the filters was widely used in previous studies (Abbott et al., 2014; Cory et al., 2013) since it can eliminate most of the microbial populations (e.g., prokaryotes and fungi) in the water (Supplementary table 2)*” (Pages 15-16, line 322-327) and “*Meanwhile, the remaining water sample for each light-exposed or dark control replicate was transferred into a 50 mL amber jar with an airtight lid after 12-hour placement, during which ROS was assumed to be largely quenched and would exert limited impact on microbes (Ward et al., 2017)*” (Page 16, line 337-340).

Table R2. Microbial cell counts measurements for Milli-Q water, GF/F filtered and 0.22 μm filtered thermokarst lake water and the latter exposed to simulated sunlight or kept in the dark.

Lake Name	Filter	Experiment Treatment	Microbial cell counts (× 10 ⁴ cells mL ⁻¹)
NQ	GF/F	/	11
NQ	0.22 μm	/	3
NQ	0.22 μm	Light-exposed 21 hours	3
NQ	0.22 μm	Dark control 21 hours	10

SN	GF/F	/	25
SN	0.22 μm	/	1.7
SN	0.22 μm	Light-exposed 21 hours	2
SN	0.22 μm	Dark control 12 hours	4
GL	GF/F	/	64
GL	0.22 μm	/	1.7
GL	0.22 μm	Light-exposed 21 hours	5
GL	0.22 μm	Dark control 21 hours	21
Milli-Q water	/	/	5.4
Milli-Q water	/	/	4.9
Milli-Q water	/	/	3.7

Notes: the sampling sites are Nagqu (NQ), Seni District (SN) and Golog Tibetan Autonomous Prefecture (GL). Microbial cell counts are determined by flow cytometry (Agilent NovoCyte 1040).

[Comment 10] Line 174: *DOM content is not related to the chemistry of photo-degraded compounds? What does this statement mean? That there was no correlation between the concentration of DOM (i.e., DOC) and any of the mass spec data of the photo-degraded compounds? Because FTICR mass spec data is not quantitative (e.g., Hockaday et al. 2009 in Limnology and Oceanography, Sleighter et al. 2012 in Analytical Chemistry, among many other papers), one would not expect a correlation between concentration (on a mass basis such as DOC) and mass spec signals.*

[Response] Sorry for the confusion. We do agree with the reviewer that FT-ICR MS spec data is not quantitative and it could not be used to quantify the concentration of different molecular compounds. The aim of this correlation analysis was to explore whether the chemical characterization (represented by the H/C ratio, AI_{mod} , DBE, NOSC, and so on) of the photo-produced and photo-degraded compounds would change with DOM concentration. To avoid confusion, we have deleted the sentences in the revised MS. Thanks for your understanding!

[Comment 11] Line 191-193: The author’s data do contrast the traditional view, as do the papers cited in this section of the manuscript. Thus, the authors could more accurately describe their contribution as building on the newer view of sunlight’s effects on DOM photodegradation by reproducing the same observations as others (e.g., Ward et al. 2017 in Nature Communications, Nalven et al. 2020 in Env. Micro., Bowen et al. 2019 in Limnology & Oceanography) but in a different region (e.g., thermokarst lakes vs. arctic or temperate freshwaters in these other studies).

[Response] Very good suggestion! Following the reviewer’s suggestion, we have rephrased these sentences to describe our contribution more accurately as follows: *“Taken together, in support of the emerging view (Bowen et al., 2020b; Nalven et al., 2020; Ward et al., 2017), but contradicting the traditional view (Amado et al., 2007; Tranvik et al., 2001; Wetzel et al., 1995) in arctic and temperate ecosystems, our results demonstrated that the promotion of microbial degradation by sunlight increased with the aromaticity of photo-produced compounds in thermokarst lakes” (Page 10, line 197-200).*

[Comment 12] Line 210: “the enrichment of these taxa” is confusing, in what lake or treatment where these taxa enriched? What is the baseline for comparison here?

[Response] Sorry for the confusion caused by our inaccurate description. We did not mean to compare the abundance of these taxa among lakes or treatments. Actually, we just aimed to depict the overall profile of the microbial composition and function in these lakes, especially the dominant taxa and their corresponding carbon degradation function. To avoid misleading, and combine suggestions from the second reviewer (*“the authors analyzed microbial diversity of the inocula used for the experiments and not the experimental incubations, but in several places the text uses the word “enriched” making it sound like the results are from the experimental incubations”*), **we have deleted these sentences in the revised MS.**

[Comment 13] Line 213-225 and Figure 3: This section and figure is confusing. What is being compared here? The relative abundance of microbes that break down aromatics vs cellulose? Is the comparison across the 10 thermokarst lakes, or between light and dark treatments? It's not clear that there is a light vs. dark control difference here in microbial community composition or function, so it is not clear how these data support (or do not support) the effect of sunlight on microbial respiration of DOC.

[Response] Sorry for the confusion. As mentioned above, we did not compare the abundance of these taxa among lakes or treatments. **What we actually compared was the relative abundance of different microbial taxa by taking 10 thermokarst lakes as replicates.** By doing so, we aimed to depict the overall profile of the microbial composition and function in these lakes, especially the dominant taxa and their corresponding carbon degradation function. For example, Fig. R2a revealed an overall higher relative abundance of prokaryotes degrading aromatic compounds than those degrading cellulose in the thermokarst lakes. Likewise, Fig. R2b revealed that compared to other fungal trophic modes, saprophytic fungi that are strongly associated with aromatic compound degradation (Grossart et al., 2019) were the most abundant fungal groups in the thermokarst lakes. Taken together, these results of microbial functional prediction indicated that the microbes in Tibetan thermokarst lakes were actively involved in the degradation of aromatic DOM. The overall microbial functions could partly explain why aromatic-like compounds rather than aliphatic-like compounds produced by sunlight promote microbial degradation. To make it clearer, we have added these explanations in the revised MS as follows: “*To test this possibility, we first used high-throughput sequencing (prokaryotic 16S rRNA gene, fungal ITS rRNA gene) to elucidate the overall profile of the microbial composition in thermokarst lakes*” (Page 10, line 207-209) and “*To further reveal the potential functions of the microbial community in the thermokarst lakes, FAPROTAX (Louca et al., 2016) and FUNGuild (Nguyen et al., 2016) were used to predict prokaryotic and fungal functions based on Operational Taxonomic Unit (OTU) taxonomic information, respectively. The analysis showed that the relative abundances of prokaryotes involved in aromatic*

compounds degradation were significantly higher than those associated with cellulose degradation in the thermokarst lakes (Fig. 3c). Likewise, saprophytic fungi were the most abundant functional groups (Fig. 3d). This fungal group has been reported to be strongly associated with aromatic compound degradation (Grossart et al., 2019). All of these findings were in accord with a recent study indicating that the taxa and copiotrophs in Tibetan lakes were actively involved in the degradation of terrestrially-derived DOC (Yang et al., 2020)” (Page 11, line 216-225). Thanks for your understanding!

Fig. R2. Predicted function of microbial communities across lakes. (a-b) Relative abundance (%) of prokaryotes degrading cellulose and aromatic compounds in FAPROTAX function prediction (a) and fungal trophic modes after FUNGuild analysis (b). Significant differences are denoted by different letters ($P < 0.05$). Patho: Pathotroph, Sapro: Saprotroph.

[Comment 14] Line 256: It is not accurate to say that this manuscript uncovered a mechanism of response of DOC biodegradation in response to sunlight given that the paragraph above (lines 239-253) reports nearly identical results to published literature (e.g., Ward et al. 2017 in Nature Communications). A more accurate summary would be to say that this manuscript here builds on prior work by showing that this newer view of photo and biodegradation of DOC appears to explain DOC degradation in thermokarst lakes on the Tibetan Plateau.

[Response] Following the reviewer’s suggestions, we have rephrased the summary as

follows: “*In summary, consistent with the emergent view (Bowen et al., 2020b; Nalven et al., 2020; Ward et al., 2017) but contrasted with the traditional view (Amado et al., 2007; Tranvik et al., 2001; Wetzel et al., 1995) found in arctic and temperate ecosystems, our results demonstrated that the enhancement of sunlight on DOC biodegradation in thermokarst lakes across the Tibetan Plateau was governed by the more oxidized and aromatic-like compounds rather than the low molecular weight, aliphatic-like compounds produced by sunlight (Fig. 5)*” (Page 13, line 263-268).

[Comment 15] Line 268: Please revise “more complicated than originally anticipated”. There is no paper suggesting that DOC degradation is “not complicated” (and it is not possible to define “complicated”).

[Response] Sorry for the inappropriate description. Following the reviewer’s comment, we have deleted this sentence in the revised MS (Page 13, line 271-274).

[Comment 16] Lines 256-279: Can the authors do more to put their results into context of regional carbon budgets? For example, how significant is the potential for photo-bio degradation of DOM to the CO₂ released from these lakes? Given that the lakes are generally located in swamp meadows, is there any shading of lake water by vegetation that would limit the importance of photo-bio degradation? The authors start out with a quantitative argument in the introduction (lines 39-57), thus the manuscript reads as incomplete by not returning to the quantitative justification for the work in the conclusions and implications section.

[Response] Very good comment! As mentioned above, quantitative analysis for the photo-biodegradation of DOC as a source of CO₂ in these lakes was not the focus of this MS. Therefore, following the reviewer’s suggestions, we have weakened the quantitative argument in the *Introduction* session to remove the quantitative justification (Pages 2-3, line 33-48). Meanwhile, we have set up the knowledge gap as not knowing which of the two alternative views would apply in the thermokarst lakes. Furthermore, based on the specificities of the DOC properties in thermokarst lakes, we

have also proposed a hypothesis and discussed why the emerging view from Ward et al. (2017) and Nalven et al. (2020) would be expected to apply in the thermokarst lakes (Page 4, line 65-77).

Regarding the potential shading by plants, we would like to clarify that although the lakes are generally located in swamp meadows and alpine meadows, limited vegetation grows in the water as depicted in Fig. R3. Thus, no vegetation shading will affect the importance of photo-bio degradation in our study.

Fig. R3. Aerial and close-up images of thermokarst lakes on the Tibetan plateau. Aerial images (left) and close-up images (right) of thermokarst lakes in Madoi County (MD, a) and Maqên County (MQ, b) in swamp meadows and alpine meadows.

[Comment 17] Conceptual figure 5: What is the cause of the “increase of DOM content and terrestrially-derived DOC”? Please note inconsistent use of DOC and DOM, molecule and compound throughout the manuscript. It is also not clear what is the difference between the left and right panel in this figure? I think the left panel is

indicating less CDOM (i.e., less aromatic DOM), and thus less UV enhancement of DOM biodegradation compared to the right panel? How many lakes in the region fall into these two categories?

[Response] Sorry for the confusion. Before answering the reviewer's comments, we would like first to clarify the meaning of conceptual Figure 5. What this figure aims to reflect is the regional pattern of UV light effect on microbial degradation of DOC across the Tibetan thermokarst lakes, that is, with the increase of DOM and CDOM content (from the left to the right), the enhancement of light on the biodegradation of DOC in thermokarst lakes also intensifies (from the left to the right). Thus, the left and right panels below only showed the states at both ends of the gradient, rather than the two split states. In contrast to the lakes in the right panel, the lakes in the left panel have less CDOM (*i.e.*, less aromatic DOM). As a result, the sunlight effects on DOC biodegradation were less in these lakes. Since these two panels are in a state of gradual change, we cannot accurately classify different thermokarst lakes into one of them. To make it clearer, we have revised the conceptual figure in the revised MS (Fig. R4).

Regarding the drivers of the “increase of DOM content and terrestrially-derived DOC” across the gradient, this could be attributed to the increase in vegetation NPP and soil organic carbon content across this gradient (Ma et al., 2019). As for the inconsistent use of DOC and DOM, molecule and compound throughout the manuscript, following the reviewer's comment, we have modified all the terms “DOM” and “molecule” to “DOC” and “compound”, respectively. Nevertheless, we would like to clarify that the DOM content used in Fig.1 and Fig.5 represented the first component of PCA analysis for DOC and DON content, which was different from DOC. Thus, the word “DOM” in Fig.1 and Fig.5 and the related description cannot be united with “DOC”. Thanks for your understanding!

Fig. R4. Conceptual diagram of the mechanism of UV light effect on microbial degradation of DOC in thermokarst lakes. With the increase of DOM content (represented as the PC1 of PCA for DOC and DON content) and terrestrially-derived DOC (from the left to the right), more small aromatic-like compounds will be produced by sunlight exposure. Given that the microbial communities in thermokarst lakes have the great potential ability to decompose aromatic compounds, thermokarst lakes with more increase in aromatic-like compounds after UV exposure will exhibit larger enhancement of sunlight on subsequent microbial degradation of DOC.

[Comment 18] Overall, it is not clear what the CO₂-RR (response ratio) is? Line 100-101 indicates it is a ratio between the sunlight treatment and the “control group” but the methods section does not clarify what is the control group?

[Response] Sorry for the confusion. The CO₂-RR (response ratio) is the ratio of microbial CO₂ production between the sunlight treatment and the dark control. To avoid confusion, we have modified the “control group” to the “dark control” (Page 5, line 96-98).

[Comment 19] Line 295: Was the GF/C filtered inoculum frozen at -20 °C prior to the incubation? If so, what effect did that storage method have on the microbial

communities?

[Response] Yes, the GF/C filtered inocula were frozen at -20°C during the transportation. This storage method was chosen due to the large sampling scale (*i.e.*, 123 thermokarst lakes along an 800-km permafrost transect) with a relatively long sampling duration (*i.e.*, from July to August 2020) in our study. Given that the freezing temperature at -20°C was generally considered adequate to store samples for microbial analysis (Brandt et al., 2014; Delavaux et al., 2020), this storage method was chosen to keep the microbes in thermokarst lakes in their original state during this long-term sampling. Nevertheless, we acknowledged that this freezing storage would inevitably reduce microbial activities, which might potentially affect the subsequent biodegradation experiment. Therefore, to minimize this potential effect, pre-incubation at 20°C was conducted for the inoculum to activate microbes and avoid pulses in microbial activities induced by changing temperature. This combination of freezing storage and pre-incubation has been widely used in previous large-scale soil incubations (Bai et al., 2020; Bastida et al., 2019; Rijkers et al., 2022). Following the reviewer’s comment, we have added a short paragraph to discuss the potential effects of this storage method in the revised MS as follows: *“Although this freezing temperature was generally considered adequate to store samples for microbial analysis (Brandt et al., 2014), this storage would inevitably reduce microbial activities. Therefore, to minimize this potential effect, pre-incubation at 20°C was conducted for the inocula to activate the microbes and avoid pulses in microbial activities induced by changing temperature before the microbial degradation experiment (Bai et al., 2020; Bastida et al., 2019; Rijkers et al., 2022)”* (Page 15, line 309-314). Thanks for your understanding!

[Comment 20] Line 307: *What was the time between sample collection and filtration and the start of the incubation experiment? I.e., how long were samples stored at -20°C? Were the samples thawed prior to the start of the incubation and how long did that take? Was it done in the dark? What was the temperature during the UV light exposure*

treatment and did this temperature differ from the dark control water?

[Response] The water samples were frozen until the UV light experiment was set up within 4 to 6 weeks. All the samples were thawed before the start of the UV light experiment in the dark at room temperature (20°C) for around 10 hours. Afterwards, the UV light experiment (*i.e.*, both the UV exposure treatment and dark control) was conducted at room temperature at around 20°C. We have clearly mentioned these points in the revised MS as follows: *“The water samples were frozen until the UV light experiment was set up within 4 to 6 weeks. All the samples were thawed before the start of the UV light experiment in the dark at room temperature (20°C) for around 10 hours. Then, the UV light experiment (i.e., both the UV exposure treatment and dark control) was conducted at room temperature”* (Page 15, line 318-322).

[Comment 21] Line 314: *At what wavelength(s) is the spectral intensity compared between the UVA bulbs and the solar irradiance reaching the Tibetan Plateau? 340 nm? Is the 9 hour comparison here saying that 21 hours of experimental light at 340 nm would take 9 hours for the same 340 nm light to reach the surface of the lakes? If so, this is a helpful but incomplete comparison. A more rigorous and helpful comparison would be to compare the rate of light absorbed by the CDOM in the lake vs. the experiment (see Vahatalo et al. 2000 in Limnology and Oceanography, Vahatalo and Wetzel 2008 in Limnology and Oceanography, Cory et al. 2014 in Science). Given that the authors have the CDOM data as reported in Supplementary Dataset 1, they can make these calculations.*

[Response] Good comment! This comment enables us to dig into the comparison of solar irradiance. The data of solar radiation on the Tibetan plateau used in the original MS was the total surface solar radiation intensity from 200-4000 nm with the unit of W m⁻². However, our measured data of the UVA-340 lamp was the total radiation intensity in the wavelength ranging from 315 to 400 nm, with a central wavelength of 340 nm. **Given that the wavelength range of the solar radiation from the Tibetan plateau and UVA lamps does not coincide, the previous comparison was not appropriate.**

Therefore, to avoid misleading, we have deleted the related sentences in the revised MS. Moreover, following the reviewer's suggestion, we have calculated the rates of light absorption by CDOM in the 10 thermokarst lakes based on the NREL SMARTS model, where we assumed that the path length of light (z) corresponds to the average depth (0.1 m) sampled in field lakes. Meanwhile, we have also measured the UVA lamp irradiance spectra with a high-precision fibre-optics spectrometer (Maya2000 Pro, Ocean Optics). Equation (1) was used to calculate the rates of UVA absorption by CDOM in the simulated light experiment, where λ_{\min} and λ_{\max} are the minimum and maximum wavelengths of light absorbed by CDOM (315 nm and 400 nm, respectively). We assumed that the path length of light (z) was equivalent to the height of the water (0.04 m) in the experiment. **In the 315-400 nm range, the 21-hour simulated light experiment is equivalent to around 7 hours (6.68 ± 0.25) of in-situ irradiance and about 4 hours (3.68 ± 0.52) of in-situ light absorbed by CDOM in thermokarst lakes on the Tibet Plateau** (Fig. R5, Table R3). Nevertheless, as mentioned above, given the large uncertainties induced by the input data in the model prediction, we preferred not to add these results to the main text. If the reviewer insists, we could add it in the next round of revision. Thanks for your understanding!

Fig. R5. Comparison of the irradiance and the light absorbed by CDOM between simulated sunlight and natural sunlight in the field. The irradiance (a) and the light absorbed by CDOM (b) for UVA light experiments and in situ light are shown in pink and blue, respectively. Solid lines represent mean curves, shaded areas represent means \pm standard deviation.

Table R3. Comparison of simulated sunlight irradiance and light absorbed by CDOM with natural sunlight in the field.

Sites	Light exposure time (h)	Irradiance in the experiment (W m^{-2})	Irradiance in the field (W m^{-2})	Equivalent light exposure time in irradiance (h)	Light absorbed by CDOM in the experiment (mol photon m^{-2})	Light absorbed by CDOM in the field (mol photons $\text{m}^{-2} \text{h}^{-1}$)	Equivalent light exposure time in light absorbed by CDOM (h)
YS	21	15.00 ± 0.31	45.75	6.89	0.35 ± 0.01	0.12	2.92
GM	21	15.00 ± 0.31	46.29	6.80	0.82 ± 0.02	0.24	3.42
NQ	21	15.00 ± 0.31	48.89	6.44	0.69 ± 0.01	0.22	3.14
SN	21	15.00 ± 0.31	49.28	6.39	2.03 ± 0.04	0.46	4.41
NR	21	15.00 ± 0.31	49.67	6.34	1.22 ± 0.02	0.35	3.49
AD	21	15.00 ± 0.31	49.18	6.41	1.72 ± 0.04	0.43	4.00
GL	21	15.00 ± 0.31	45.64	6.90	1.18 ± 0.02	0.31	3.81
MD	21	15.00 ± 0.31	46.11	6.83	1.35 ± 0.03	0.34	3.97
HH	21	15.00 ± 0.31	45.02	7.00	0.55 ± 0.01	0.17	3.24

MQ	21	15.00 ± 0.31	46.13	6.83	1.71 ± 0.03	0.39	4.38
----	----	--------------	-------	------	-------------	------	------

Notes: the ten sampling sites are Yushu (YS), Golmud (GM), Nagqu (NQ), Amdo County (AD), Nyainrong County (NR), Seni District (SN), Heihe River (HH), Golog Tibetan Autonomous Prefecture (GL), Madoi County (MD), and Maqên County (MQ).

[Comment 22] Line 325: ROS quenching was assumed to take place over 12 hours but ROS were not measured, correct? Please clarify what was measured vs. assumed.

[Response] No, we did not measure ROS in this study. ROS quenching was assumed to take place over 12 hours based on a recent study (Ward et al., 2017). Following the reviewer's comment, we have clarified this point in the revised MS as follows: "*the remaining water sample for each light-exposed or dark control replicate was transferred into a 50 mL amber jar with an airtight lid after 12-hour placement, during which ROS was assumed to be largely quenched and would exert limited impact on microbes (Ward et al., 2017)*" (Page 16, line 337-340).

[Comment 23] Line 326: Why was the inoculum pre-incubated at 20 °C for 7-14 days prior to the experimental incubation? Can the authors discuss the potential bottle effects that change microbial communities and their activities over this time?

[Response] As mentioned above, the inocula were stored frozen until the UV light experiment was set up. This freezing storage would inevitably reduce microbial activities, which might potentially affect the subsequent biodegradation experiment. Therefore, to minimize this potential effect, **pre-incubation at 20°C was conducted for the inoculum to activate microbes and avoid pulses in microbial activities induced by changing temperature. This combination of freezing storage and pre-incubation has been widely used in previous soil incubation** (Bai et al., 2020; Bastida et al., 2019; Rijkers et al., 2022). Nevertheless, we acknowledged that this pre-incubation may cause potential "bottle effects" on microbial community composition due to time spent in the artificial environment (Hammes et al., 2010; Hartzog et al., 2017; Pound et al., 2022). Fortunately, **these potential bottle effects would not influence the treatment comparisons for microbial CO₂ production because both the light-exposed and dark control water samples received the same inoculum from the corresponding lake.** Following the reviewer's comment, we have added these points in the revised MS as follows: "*Afterwards, the inoculum (i.e., GF/C filtered water from the lake corresponding to the light-exposed and dark control) of*

approximately 6% of the sample volume was inoculated, which was pre-incubated (20 °C) to activate microbes and avoid pulses in microbial activities induced by changing temperature (Bai et al., 2020; Bastida et al., 2019; Rijkers et al., 2022). This combination of freezing storage and pre-incubation has been widely used in previous large-scale soil incubations (Bai et al., 2020; Bastida et al., 2019; Rijkers et al., 2022). It should be noted that despite this pre-incubation may cause potential “bottle effects” on microbial community composition (Hammes et al., 2010; Hartzog et al., 2017; Pound et al., 2022), it would not influence the treatment comparisons because the water samples in the light-exposed treatment and dark control received the same inoculum from the corresponding lake” (Pages 16-17, line 340-349). Thanks for your understanding!

[Comment 24] Line 327: What was the purpose of the sterile water blanks? Were these blanks analyzed by FTICR or for respiration or some other purpose?

[Response] Sorry for the confusing description. Actually, equal volumes of sterile water were set as blank to eliminate the shift in headspace CO₂ remaining in the bottle after flushing with CO₂-free air during the incubation. Thus, only CO₂ concentration was measured for these blanks as background CO₂ at each measurement period, which was subtracted when calculating the total DIC in the bottle. We have clearly mentioned these points in the revised MS as follows: “Additionally, water samples that received equal volumes of sterile water were set as blank to eliminate the shift in headspace CO₂ remaining in the bottle after flushing with CO₂-free air during the incubation” (Page 17, line 349-351).

[Comment 25] Line 325: What was the “respective inoculum” in each sample? Was it the GF/C filtered water from the lake corresponding to the light-exposed and dark GF/F filtered water?

[Response] Yes, the respective inoculum was the GF/C filtered water from the lake corresponding to the light-exposed and dark control. We have clearly mentioned these

points in the revised MS as follows: “Afterwards, the inoculum (i.e., GF/C filtered water from the lake corresponding to the light-exposed and dark control) of approximately 6% of the sample volume was inoculated, which was pre-incubated (20°C) to activate microbes and avoid pulses in microbial activities induced by changing temperature (Bai et al., 2020; Bastida et al., 2019; Rijkers et al., 2022)” (Page 16, line 340-343).

[Comment 26] Line 329: Is there data to support that the flushing with CO₂ free air result in similar initial CO₂ concentrations in the water at the start of the incubation?

[Response] Sorry for the inaccurate description in the original MS. Actually, flushing with CO₂-free air could only homogenize headspace CO₂ concentration in the incubation bottle (Dean et al., 2019; Ding et al., 2016), but not the initial CO₂ concentration in the water, because there are large variations in the DIC concentration in different water samples. Based on our preliminary experiment, the initial headspace CO₂ concentrations after flushing with CO₂-free air within the bottles were similar, ranging from 29.87 to 60.85 ppm, with a standard error of 10.01 ppm (Fig. R6). We have clearly mentioned these points and added the results in the revised MS as follows: “After pre-incubation, all the water samples were flushed with CO₂-free air for 10 min to homogenize the initial headspace CO₂ concentration in the jar (Supplementary Fig. 9) (Dean et al., 2019; Ding et al., 2016)” (Page 17, line 351-353). Thanks for your understanding!

Fig. R6. The initial headspace CO₂ concentrations (ppm) of samples. The sampling sites are Nagqu (NQ), Seni District (SN), Nyainrong County (NR), Amdo County (AD), Golog Tibetan Autonomous Prefecture (GL), Madoi County (MD), Heihe River (HH), and Maqên County (MQ). B1 to B10 represent ten blank samples.

[Comment 27] Line 340-344: How is the experimental incubation different here when measuring composition of biodegraded compounds vs. the incubations to measure respiration (CO₂ in the headspace) described above? In addition, what is the control for the analysis of biodegraded DOC? That is, when doing FTICR analysis of DOM, it is very challenging to determine an effect of a treatment given the qualitative nature of this analysis combined with less than 100% extraction efficiency by PPL. In addition, there is biodegradation of DOC occurring in the inoculum which may dilute any treatment signal (e.g., the effect of light in this case). While the authors indicated no significant loss of DOC during the light-exposure portion of the experiment, differences in extraction efficiency between samples would be expected given the large changes in DOM composition, and these differences in extraction efficiency may account for a lot of difference in sample spectra analyzed by FTICR mass spec.

[Response] The incubations for measuring microbial respiration and for the measuring chemical composition of biodegraded compounds had the same settings (*i.e.*, incubation temperature, duration, and so on), but in separate bottles, because different pre-treatments are required for the two measurements (*i.e.*, respiration: add phosphoric acid to convert all the DIC into CO₂; FT-ICR MS: add hydrochloric acid to pH 2).

Regarding the analysis of biodegraded DOC, we do agree with the reviewer that it is very challenging to determine an effect of a treatment (light vs. dark here) due to the potential biases induced by FT-ICR MS analysis with less than 100% PPL extraction efficiency, different PPL extraction efficiency for different DOC chemical composition and so on. Nevertheless, **it should be clarified that we did not assess the differences in biodegraded DOC between the light treatment and the dark control. Instead,**

we only compared the chemical properties of biodegraded DOC with those of photo-degraded and photo-produced DOC in our MS. Thus, the control for the analysis of biodegraded DOC was the photo-degraded and photo-produced DOC. The potential biases induced by FT-ICR MS analysis will not affect our results. Thanks for your understanding!

[Comment 28] Line 376: What was the extraction efficiency on the PPL column? What was the reproducibility of the FTICR spectra? For example, were samples analyzed in duplicate or triplicate and if so, how many masses were common to the replicates? (See Sleighter et al. 2012 in Analytical Chemistry, Ward and Cory 2015 in GCA).

[Response] The extraction efficiency on the PPL column was 62.29 ± 11.99 % in our study (Table R4). Regarding the reproducibility of the FTICR spectra, given the high cost of FT-ICR MS, only water samples from three thermokarst lakes (NQ, SN, GL) were analyzed in duplicate to explore the reproducibility of the FTICR spectra. Our analysis showed that 72.64 ± 2.41 % of the peaks were common to the replicates of the same sample (Table R4). We have added this additional information in the *Methods* session of the revised MS as follows: “*The extraction efficiency on the PPL column was 62.29 ± 11.99 % (Supplementary Table 4)*” (Page 20, line 413-414) and “*In addition, 72.64 ± 2.41 % of the peaks were common to the replicates of the same sample (Supplementary Table 4)*” (Page 20, line 424-425). Thanks for your understanding!

Table R4. PPL extraction efficiency (%) and reproducibility (%) in thermokarst lakes.

Lakes	Treatment	PPL Extraction Efficiency (%)	Number of Common Peaks	Reproducibility (%)
NQ	Dark	61.23	5138	71.51
NQ	Light	86.00	6710	69.17
SN	Dark	56.28	6300	71.49
SN	Light	59.11	6829	73.11
GL	Dark	52.49	8694	75.72

GL	Light	58.67	9538	74.82
Mean (SD)		62.29 (11.99)	/	72.64 (2.41)

Notes: SD, standard deviation. The sampling sites are Nagqu (NQ), Seni District (SN) and Golog Tibetan Autonomous Prefecture (GL).

Overall, we greatly appreciate the reviewer’s insightful comments. As mentioned above, we have supplemented methodological details to facilitate the reader’s evaluation, interpreted the results in more depth and clarified assumptions and measurements. Additionally, we have reset the knowledge gap to avoid the quantitative justification in the *Introduction* session. By doing so, we feel that our conclusion becomes more convincing and the revised manuscript has been greatly improved. Thank you!

Responses to Reviewer #2

[Comment 1] This paper presents a study of the combined controls of sunlight exposure and microbial degradation on dissolved organic carbon mineralization in lakes formed by thermokarst failures of permafrost. These lakes vary in DOC content and serve as important conduits for the flux of carbon from land to atmosphere. The authors conducted incubation experiments in which they photo-exposed filter-sterilized DOC from a range of lakes with the equivalent of ~ one day of sunlight, and then inoculated that organic matter with bacteria from its lake of origin. They found many remarkably strong relationships between DOC chemistry and the effect of sunlight exposure on the degree to which sunlight enhanced or reduced microbial respiration. They concluded that photo-production of small aromatic compounds was the main mechanism by which sunlight enhanced microbial respiration. They also concluded that aromatic-compound degrading microbial taxa were abundant in their samples. I think this was a well-conceived, high-quality study that found very strong relationships that are potentially ground-breaking. The methodology is sound, and the methods provide adequate detail. This work will be significant for the field.

[Response] We are very grateful to the reviewer for the positive and insightful comments on our manuscript! These comments, together with those listed below enabled us to have deeper thinking on this issue and thus guided us to conduct a thorough revision of the original manuscript. For detailed modifications please see our responses to the following comments.

[Comment 2] My main concerns are that: 1. The Results/Discussion section is difficult to understand for a non-expert in aquatic organic carbon chemistry. This is in part unavoidable because of the nature of this complicated experiment. But it is also because of the huge number of abbreviations, most of which readers are expected to understand and remember.

[Response] Very good comment! Following the reviewer's comment, we have replaced the abbreviations with their original meaning throughout the manuscript, whenever

possible, to improve readability (Page 8, line 158-161; Pages 8-9, line 163-171; Page 9, line 176-183; Page 9, line 185-190; Pages 11-12, line 232-245).

[Comment 3] 2. Conclusions concerning patterns in microbial community composition and predicted functions appear to be overstated and potentially misleading. First, the authors analyzed microbial diversity of the inocula used for the experiments and not the experimental incubations, but in several places the text uses the word “enriched” making it sound like the results are from the experimental incubations. Second, many of these taxa are very common in freshwater ecosystems, but for some reason the authors describe them as “enriched” in thermokarst lakes. Third, some of the text implies that there is a relationship between potential aromatic-degrading taxa and DOM chemistry measurements. For example, in Fig. 5 they show a gradient bar labeled “Microbes that degraded aromatic compounds” right next to a similar gradient bar labeled “Light enhancement on DOC biodegradation.” I cannot find where the authors show this relationship in their results. Fourth, the abstract says DOC degradation is “attributed to the high relative abundance and diversity of bacteria and fungal taxa that have great abilities to decompose complex compounds,” but these taxa are abundant in many different freshwater systems, and so this attribution seems suspect.

[Response] Very good comment! Following the reviewer’s suggestion, we have made the following modifications:

- Regarding the term “enrichment”, we acknowledged that the use of this word was not appropriate. Actually, we had no intention to compare the microbial community composition between treatments or ecosystems. What we compared in the MS was the relative abundance of different microbial taxa by taking 10 thermokarst lakes as replicates. Thus, what we intended to convey was that compared with other microbial taxa, Burkholderiales and Ascomycota were the dominant prokaryotic and fungal groups in the thermokarst lakes across the Tibetan Plateau. To avoid confusion, we have deleted the term “enrichment” throughout the MS and rephrased the related sentences as follows: “This

aromatic compound-driven acceleration of microbial DOC degradation could be possibly attributed to the potential high abilities of the microbes to decompose complex compounds in thermokarst lakes” (Pages 5-6, line 102-104) and “Likewise, saprophytic fungi were the most abundant functional groups (Fig. 3d). This fungal group has been reported to be strongly associated with aromatic compound degradation (Grossart et al., 2019)” (Page 11, line 222-224).

- Regarding the common taxa in the freshwater ecosystems, we agreed with the reviewer that many of the taxa (*e.g.*, Burkholderiales) are very common in freshwater ecosystems. As mentioned above, what we intended to convey was that compared with other microbial taxa, Burkholderiales and Ascomycota were the dominant prokaryotic and fungal groups in the thermokarst lakes across the Tibetan Plateau. **To avoid confusion, we have deleted the term “enriched” and rephrased the sentences** as follows: “*To test this possibility, we first used high-throughput sequencing (prokaryotic 16S rRNA gene, fungal ITS rRNA gene) to elucidate the overall profile of the microbial composition in thermokarst lakes. We found that Burkholderiaceae were the most dominant family in prokaryotes (i.e., bacteria and archaea), which comprised 18% of total sequence reads, followed by Sphingomonadaceae (16%), Verrucomicrobiaceae (10%), and Flavobacteriaceae (9%) (Fig. 3a). Similarly, Eurotiomycetes and Tremellomycetes were the dominant groups at the class level in fungi, accounting for 12% and 10% respectively (Fig. 3b)*” (Pages 10-11, line 207-214).
- Regarding the text implying that there is a relationship between potential aromatic-degrading taxa and DOM chemistry measurements, **we have carefully checked the descriptions of the whole MS and removed all these arguments**, because these relationships were not significant. Specifically, we have removed the gradient bar labelled “Microbes that degraded aromatic compounds” in the conceptual figure (Fig. R7).
- Regarding the argument in the *Abstract* session about the linkage of DOC degradation and microbial taxa, we agreed with the reviewer that Burkholderiales

was the common dominant prokaryotic taxon at the order level in the global freshwater ecosystems. Therefore, we have deleted the argument about the linkage of DOC degradation and microbial composition in both *Abstract* and *Results and discussion* sessions. Nevertheless, we did provide two lines of evidence to demonstrate the potential high ability of microbes to consume aromatic compounds. **First**, FAPROTAX-based functional prediction revealed that the relative abundance of prokaryotes to break down aromatic compounds was significantly higher than those to degrade cellulose in the thermokarst lakes. Likewise, compared with other fungal groups, saprophytic fungi that were strongly associated with aromatic compound degradation (Grossart et al., 2019) were also revealed as the most abundant fungal group in the thermokarst lakes. These results from microbial functional prediction jointly suggested that the microbes in Tibetan lakes may be actively involved in the degradation of aromatic compounds. **Second**, the chemistry of DOC compounds consumed by microbes was generally with higher aromaticity (*i.e.*, AI_{mod}), higher unsaturation (*i.e.*, DBE) and less lability (*i.e.*, lower H/C ratio) than those compounds produced by sunlight. Likewise, the proportion of combustion-derived polycyclic aromatics (CA) degraded by microbes was much higher than that of highly unsaturated and phenolic compounds (Uns.) and vascular plant-derived polyphenols (Pol.) (Fig. 4a insert box plots), confirming that microbes in the thermokarst lakes tend to degrade these highly aromatic-like compounds. Therefore, based on these results, we have toned down the argument and rephrased the sentence in the *Abstract* session as follows: “*This aromatic compound-driven acceleration of biodegradation may be attributed to the potential high abilities of the microbes to decompose complex compounds in the thermokarst lakes*” (Page 2, line 26-28). Thanks for your understanding!

Fig. R7. Conceptual diagram of the mechanism of UV light effect on microbial degradation of DOC in thermokarst lakes. With the increase of DOM content (represented as the PC1 of PCA for DOC and DON content) and terrestrially-derived DOC (from the left to the right), more small aromatic-like compounds will be produced by sunlight exposure. Given that the microbial communities in thermokarst lakes have the great potential ability to decompose aromatic compounds, thermokarst lakes with more increase in aromatic-like compounds after UV exposure will exhibit larger enhancement of sunlight on subsequent microbial degradation of DOC.

[Comment 5] Abstract Lines 25-27. I see where there are positive relationships between CO₂-RR and the abundance of photoproducted aromatic compounds, but I don't see where there are negative relationships between CO₂-RR and the abundance of photoproducted aliphatic compounds. This is not clearly explained and defended in the text.

[Response] Good comment! It should be clarified that a positive relationship of CO₂-RR with the ratio of photo-produced combustion-derived polycyclic aromatics to aliphatic compounds (CA/Ali.) was shown in Fig. 2f in the original MS. Following the reviewer's comment, to further back up this argument, we also analyzed the relationships of CO₂-RR with four types of photo-produced compounds separately

in the revised MS (Fig. R8). Our additional analysis showed that CO₂-RR was negatively associated with the photo-produced aliphatic compounds, but positively correlated with the other three types of photo-produced aromatic compounds (Fig. R8).

We have added these results in the revised MS as follows: “Furthermore, these photo-produced unsaturated (i.e., higher DBE), oxidized (i.e., higher NO₃C) and aromatic-like compounds (i.e., the number of Uns., Pol., CA, and the CA/Ali. Ratio) were found to promote DOC biodegradation indicated by their positive associations with CO₂-RR (Fig. 2b-f; Supplementary Fig. 6b-c). By contrast, photo-produced aliphatic compounds were observed to inhibit DOC biodegradation indicated by its negative association with CO₂-RR (Supplementary Fig. 6a)” (Page 9, line 186-191).

Fig. R8. Relationships between microbial respiration response to UV light exposure (CO₂-RR) and the number of photo-produced compounds. Correlations of CO₂-RR with the number of photo-produced aliphatic compounds (Ali., a), highly unsaturated and phenolic compounds (Uns., b), vascular plant-derived polyphenols (Pol., c) and combustion-derived polycyclic aromatics (CA, d). The solid lines are the fitted lines obtained based on the linear model, and the grey area represents the 95% confidence interval. * $P < 0.05$, ** $P < 0.01$.

[Comment 6] Lines 28-30. These appear to be typical freshwater taxa, and it is not clear why the authors would describe their abundances as “relatively high.”

[Response] Sorry for the confusion! As mentioned above, we did not intend to compare the differences in these taxa between thermokarst lakes and other freshwater ecosystems. According to our results, Burkholderiales was the most dominant order in prokaryotes, and Ascomycota was the dominant group at the phylum level in fungi

across the thermokarst lakes. Thus, compared with other microbial taxa in the thermokarst lakes, we described these taxa as the dominant prokaryotic and fungal groups with high “relative abundance”. To avoid misleading, following the reviewer’s comment, we have deleted the argument about the linkage of DOC degradation and microbial composition in both *Abstract* and *Results and discussion* sessions. Additionally, we have toned down the argument in the *Abstract* session as follows: “*This aromatic compound-driven acceleration of biodegradation may be attributed to the potential high abilities of the microbes to decompose complex compounds in thermokarst lakes*” (Page 2, line 26-28). Thanks for your understanding!

[Comment 7] Line 31. I thought FT-ICR MS only effectively captures larger molecular weight compounds, and that small compounds such as sugars and amino acids are not easily detectable. I think it is important that the authors define what they mean by “small” compounds here.

[Response] We agree with the reviewer that FT-ICR MS is more capable of capturing large molecule compounds than small compounds. Actually, the term “small” in the original manuscript was not referred to small compounds such as sugar and amino acids. What we intended to convey was that sunlight degrades large aromatics into smaller aromatics. To avoid this confusion, we have deleted the term “small” in the revised MS. Thanks for your understanding!

[Comment 8] Line 39. I don’t understand the odd superscript and subscript for the number 1014.

[Response] The superscript and subscript for the number 1014 are the lower and upper prediction intervals at a 95% confidence level of uncertainty. For ease of understanding, we have removed the number in the revised MS.

[Comment 9] Lines 137-140. I don’t understand this sentence.

[Response] Sorry for the unclear expressions in the original manuscript. Actually, **to**

rule out the role of microbes during the UV degradation, we isolated the water samples with sterile 0.22 µm filters (Sterivex GP 0.22, Millipore) before light exposure. This pore size of the filters was widely used in previous studies assuming that it can eliminate most of the microbial populations (*e.g.*, bacteria and fungi) in the water (Abbott et al., 2014; Cory et al., 2013). Thus, the changes in microbes after the sunlight experiment would barely affect DOC biodegradation.

Regarding the role of ROS, **after the sunlight exposure experiment, the water samples were placed for 12 hours before being used for subsequent microbial degradation experiments. Based on the previous studies (Ward et al., 2017), the ROS could largely decay during this 12-hour placement.** Thus, we assumed that the influence of ROS on microbes was also minimal. To make it clearer, we have added this additional information in the revised MS as follows: “*Firstly, given that most microbes were eliminated by filtering the water samples through 0.22 µm filters before the sunlight experiment (Supplementary Table 2), sunlight exposure would barely affect the microbes. Additionally, we assumed that the reactive oxygen species (ROS) could largely decay during the 12-h placement after the sunlight experiment (Ward et al., 2017), and its impact on biodegradation was also minimal in our study (See Methods for details)*” (Page 7, line 135-141).

[Comment 10] Lines 141-144. Supplementary figures 3a-d shows some significant changes in DIC, DON, and pH in some lakes following sunlight exposure. It seems incorrect to say that sunlight did not affect these measurements.

[Response] We would like to clarify that the error bars on the dots represented standard errors (SE) rather than standard deviation (SD) in Supplementary Figures 3a-d. To avoid confusion, we have added the related information in the figure legends. Actually, the statistical significance of changes in DIC, DON, and pH had been indicated by the asterisk for each lake. As shown in the figures (Fig. R9), only one lake (AD) had a significant change in pH following light exposure, while the others had no significant

change in DOC, DIC, DON and pH after sunlight exposure. Hence, we think that it is reasonable to say that sunlight did not affect these measurements. Thanks for your understanding!

Fig. R9. Response ratios of water properties to sunlight exposure in the ten thermokarst lakes. The results of the two-way ANOVA of sunlight and lake location on water properties are shown in the lower-left corner. The error bars on the dots represented standard errors (SE). The inserted box plot depicts the main effect of sunlight exposure on water properties. The ends of the boxes represent the 25th and 75th percentiles. The horizontal lines inside each box and the whiskers show the mean and 1.5 times the standard deviation. The black dashed line denotes the response ratio of zero. * $P < 0.05$; ** $P < 0.01$; ns, not significant. See Supplementary Fig. 1 and Supplementary Fig. 2 for abbreviations.

[Comment II] Lines 162-167. This seems like an important point, but it is very difficult to understand, in part because the sentence includes 11 abbreviations.

[Response] For ease and clarity of understanding, we have revised the statement as follows: “the enhancement of sunlight on DOC biodegradation was not driven by the photo-produced aliphatic-like compounds, but associated with the photo-produced aromatic-like compounds, as evidenced by the CO_2 -RR decreasing with smaller (i.e., lower molecular weight, higher S_R) and more aliphatic (i.e., higher H/C) DOC, but

increasing with more unsaturated (i.e., higher DBE), more oxidized (i.e., higher the nominal oxidation state of carbon (NOSC)) DOC and higher proportions of the aromatic-like compound (i.e., higher ratios of Uns./Ali., Pol./Ali., CA/Ali.) after sunlight exposure (Fig. 1c-g)” (Pages 8-9, line 163-171).

[Comment 12] Lines 176-178. The word “specifically” suggested this sentence was going to describe details of the relationships described in the previous sentence about Fig. 2a (chemistry measurements vs. “DOM chemistry-RR”), but instead it is about something else. I don’t understand what the authors are implying here.

[Response] Sorry for the confusion. Actually, the word “specifically” was followed by the sentence “At the same time, the DOM content was not related to the chemistry of the photo-degraded compounds, but had strong links to the chemistry of photo-produced compounds (Supplementary Fig. 5a)” rather than the sentence of “the sunlight effect on DOC chemistry was more attributed to photo-produced compounds than photo-degraded compounds”. Therefore, the word “specifically” suggested the following sentence was going to describe the details of the relationship in Supplementary Figure 5a (DOM content vs. the chemistry of photo-produced compounds). Nevertheless, to avoid confusion, we have changed the conjunction “At the same time” to “Moreover” to distinguish the two sentences in the revised MS (Page 9, line 176).

[Comment 13] Line 184. This sentence implies that DBE (double bond equivalence) and NOSC (nominal oxidation state of carbon) are measurements of small aromatics, but I don’t think they are.

[Response] Sorry for the confusion. The DBE (double bond equivalents) and NOSC (nominal oxidation state of carbon) here are the measurements of all photo-produced compounds. To avoid this confusion, we have rephrased the sentence as follows: “these photo-produced unsaturated (i.e., higher DBE), oxidized (i.e., higher NOSC) and aromatic-like compounds (i.e., the number of Uns., Pol., CA, and the CA/Ali. Ratio)

were found to promote DOC biodegradation indicated by their positive associations with CO₂-RR (Fig. 2b-f; Supplementary Fig. 6b-c)” (Page 9, line 185-190).

[Comment 14] Lines 199-200. I don't understand this sentence. It seems to assume that smaller and less aromatic DOC compounds are always less abundant than larger and more aromatic DOC compounds. Is that true?

[Response] Sorry for the confusion. Actually, the abundance of different DOC components varies significantly with ecosystems. For example, as described by Ward et al. (2017), the smaller and less aromatic DOC compounds were less abundant than the larger and more aromatic DOC compounds for active layer soil, but the opposite is the case for permafrost soil (Ward et al., 2017). To avoid this confusion, we have revised the sentence as follows: “It has been proposed that microbes may have the intrinsic capability to degrade the most abundant DOC in the environment regardless of their molecular weight and aromaticity (Ward et al., 2017)” (Page 10, line 205-207).

[Comment 15] Lines 204-209. Burkholderiales are dominant members of most freshwater bacterioplankton communities, and the other taxa listed here are also common bacterioplankton. I don't understand how these taxa could be described as “enriched” in thermokarst lakes. What are the authors comparing the abundances to?

[Response] As mentioned above, we originally wanted to express that compared with other microbial taxa, Burkholderiales and Ascomycota were the dominant prokaryotic and fungal groups in the thermokarst lakes across the Tibetan Plateau. **To avoid confusion, we have removed the term “enrichment” throughout the MS and rephrased the related sentences** as follows: “To test this possibility, we first used high-throughput sequencing (prokaryotic 16S rRNA gene, fungal ITS rRNA gene) to elucidate the overall profile of the microbial composition in thermokarst lakes. We found that Burkholderiaceae were the most dominant family in prokaryotes (i.e., bacteria and archaea), which comprised 18% of total sequence reads, followed by Sphingomonadaceae (16%), Verrucomicrobiaceae (10%), and Flavobacteriaceae (9%)

(Fig. 3a). Similarly, Eurotiomycetes and Tremellomycetes were the dominant groups at the class level in fungi, accounting for 12% and 10% respectively (Fig. 3b)” (Pages 10-11, line 207-214).

[Comment 16] Lines 229-231. I don't understand this sentence.

[Response] Sorry for the confusion. Here, to further reveal the characteristics of the biological processing of DOC, we compared the overall chemistry of compounds that are consumed by microbes (*i.e.*, bio-degraded compounds) with those removed and produced by sunlight (*i.e.*, photo-degraded and photo-produced compounds) across the 10 lakes. Our analysis showed that **the chemistry of compounds consumed by microbes was generally more comparable to the compounds removed by sunlight, with higher aromaticity (*i.e.*, AI_{mod}), higher unsaturation (*i.e.*, DBE) and less lability (*i.e.*, lower H/C ratio) than those compounds produced by sunlight** (Fig. R10). These results inferred that the ability of microbes to degrade the recalcitrant DOC with high aromaticity and unsaturation was similar to the photodegradation in the thermokarst lake. We have clearly mentioned these points in the revised MS as follows: “Our analysis showed that **the chemistry of compounds consumed by microbes (*i.e.*, bio-degraded DOC) was generally more comparable to the compounds removed by sunlight (*i.e.*, photo-degraded DOC), but had higher aromaticity (*i.e.*, AI_{mod}), higher unsaturation (*i.e.*, DBE) and less lability (*i.e.*, lower H/C ratio) than those compounds produced by sunlight (*i.e.*, photo-degraded DOC)** (Fig. 4; Supplementary Fig. 8). *These results inferred that the ability of microbes to degrade the recalcitrant DOC with high aromaticity and unsaturation was similar to the photodegradation in the thermokarst lakes*” (Pages 11-12, line 232-240).

Fig. R10. Chemical properties of photo-produced (PP), photo-degraded (PD), and bio-degraded (BD) DOM. (a-c) Aromaticity index (AI_{mod}) (a), double bond equivalence (DBE) (b) and H/C ratio (H/C) (c) of PP, PD, and BD across 10 lakes. Insert box plots in panels (a-c) to display the comparison of AI_{mod} (a), DBE (b) and H/C (c) of PP, PD, and BD. The whiskers illustrate the 5th and 95th percentiles, and the ends of the boxes represent the 25th and 75th quartiles (interquartile range). The horizontal lines inside each box show the mean. Significant differences are denoted by different lowercase letters ($P < 0.05$).

[Comment 17] Lines 233-235. I don't understand what the authors mean here or why it is relevant.

[Response] As mentioned above, to reveal the characteristics of the biological processing of DOC, we explored the overall chemical properties of the compounds that are consumed by microbes (*i.e.*, bio-degraded compounds) across the 10 thermokarst lakes. Here, we further classified these compounds into four categories with decreasing aromaticity and increasing lability: combustion-derived polycyclic aromatics (CA), vascular plant-derived polyphenols (Pol.), highly unsaturated and phenolic compounds (Uns.) and aliphatic compounds (Ali.). By comparing the relative abundance of these four types of bio-degraded compounds, we can reveal which types of compounds the microbes may tend to degrade. Our analysis showed that the proportion of CA degraded by microbes was much higher than that of Uns. and Pol., and was similar to the Ali. (Fig. 4a insert box plots). Thus, these results suggested that microbes in the thermokarst lakes may tend to degrade these highly aromatic-like compounds. Thanks for your understanding!

[Comment 18] Lines 261-264. I think the gradient line on Fig. 5 labeled “Microbes that degraded aromatic compounds” is misleading. The authors did not show a relationship between the abundance of these microbes and the degree of light enhancement on DOC biodegradation.

[Response] We do agree with the reviewer's comment. To avoid misleading, we have removed the gradient bar labelled “Microbes that degraded aromatic compounds”.

[Comment 19] Line 326. Why were microbes “pre-incubated”? Is it more appropriate to say they were “stored” for 7 days while the DOC was processed? If there was a scientific justification for this “pre-incubation” then it should be presented here.

[Response] The reason for the pre-incubation of inoculums at 20°C for 7 days was to activate microbes and avoid pulses in microbial activities induced by changing temperature. This combination of freezing storage and pre-incubation has been

widely used in previous large-scale soil incubations (Bai et al., 2020; Bastida et al., 2019; Rijkers et al., 2022). Following the reviewer's comment, we have added a short paragraph to backup this processing in the revised MS as follows: "*Afterwards, the inoculum (i.e., GF/C filtered water from the lake corresponding to the light-exposed and dark control) of approximately 6% of the sample volume was inoculated, which was pre-incubated (20°C) to activate microbes and avoid pulses in microbial activities induced by changing temperature (Bai et al., 2020; Bastida et al., 2019; Rijkers et al., 2022). This combination of freezing storage and pre-incubation has been widely used in previous large-scale soil incubations (Bai et al., 2020; Bastida et al., 2019; Rijkers et al., 2022)*" (Pages 16-17, line 340-345). Thanks for your understanding!

[Comment 20] Line 410. Why were the microbes pre-incubated for 14 days rather than 7 days like the inocula? Is there a scientific justification for this?

[Response] Good comment! As mentioned above, the reason for the pre-incubation was to activate microbes and avoid pulses in microbial activities induced by changing temperature. Based on the previous studies, the duration of the pre-incubation was generally set at 1 to 15 days, of which 7 days was the most frequently used (Fig. R11). Here, the 14-day pre-incubated time for extracting microbial DNA was aimed to ensure the amount of DNA required for sequencing.

Nevertheless, to further explore whether this different pre-incubation time would affect the microbial properties, we compared our data (pre-incubated for 14 days) with the other collage in our groups who explored the microbial community composition after pre-incubated for 7 days. Our additional analysis showed that there were no significant differences in microbial community composition and alpha diversity between the two pre-incubation treatments (Fig. R12a, PERMANOVA, $R^2 = 0.09$, $P = 0.11$; Fig. R12b-c, Richness: $P = 0.83$, Shonnon: $P = 0.53$). Therefore, the effect of different pre-incubated times on the experimental results can be ignored. Thanks for your understanding!

Fig. R11. The frequency distribution of the pre-incubated period. The data were derived from 22 previous soil incubation studies (Bai et al., 2020; Bastida et al., 2019; Colman et al., 2013; Elberling et al., 2013; Feng et al., 2017; Fontaine et al., 2004; Fontaine et al., 2007; Jia et al., 2017; Li et al., 2018; Liu et al., 2017; Morrissey et al., 2017; Pascault et al., 2013; Razanamalala et al., 2018; Spohn et al., 2016; Su et al., 2022; Treat et al., 2014; Wagai et al., 2013; Wang et al., 2018; Xu et al., 2021; Zhang et al., 2020; Zhang et al., 2021; Zhu et al., 2021).

Fig. R12. Comparison of microbial community composition and alpha diversity between two different pre-incubated times. (a) Prokaryotic communities using the

principal coordinates analysis (PCoA) based on OTU data and permutational multivariate analysis of variance (PERMANOVA) analysis of the Bray-Curtis distance at different pre-incubated times. Circles represent a 95% confidence interval for each cluster. Points indicate individual microbial communities. Pink and blue represent 14 days of pre-incubated and 7 days of pre-incubated, respectively. (b-c) The box plot depicts the richness index (b) and Shannon index (c) for 7 days of pre-incubated and 14 days of pre-incubated. The ends of the boxes represent the 25th and 75th percentiles. The horizontal lines inside each box and the whiskers show the mean and 1.5 times the standard deviation. ns, not significant.

Overall, we greatly appreciate the reviewer's insightful comments. These comments inspired us to have a deeper thinking on both data analyses and results integration, and thus guided us to conduct a thorough revision of the original MS. To address these insightful comments, we described the microbial pre-incubated experiments in more detail, demonstrated the properties of microbial degradation compounds and their calculation methods, analyzed the relationship between CO₂-RR and different compounds, and revised the conceptual diagram to make it more clear. By doing so, we feel that our revised manuscript has been greatly improved and expect that the reviewer will be satisfied with the revised manuscript. Thank you!

References

1. Abbott, B. W., Larouche, J. R., Jones Jr., J. B., Bowden, W. B., Balsler, A. W. Elevated dissolved organic carbon biodegradability from thawing and collapsing permafrost. *J. Geophys. Res. Biogeosci.* **119**, 2049-2063 (2014).
2. Amado, A. M., Cotner, J. B., Suhett, A. L., Esteves, F. D., Bozelli, R. L., Farjalla, V. F. Contrasting interactions mediate dissolved organic matter decomposition in tropical aquatic ecosystems. *Aquat. Microb. Ecol.* **49**, 25-34 (2007).
3. Bai, Y. et al. Long-term active restoration of extremely degraded alpine grassland accelerated turnover and increased stability of soil carbon. *Glob. Chang. Biol.* **26**, 7217-7228 (2020).
4. Bastida, F. et al. Global ecological predictors of the soil priming effect. *Nat. Commun.* **10**, 3481 (2019).
5. Bowen, J., Ward, C., Kling, G., Cory, R. Arctic Amplification of Global Warming Strengthened by Sunlight Oxidation of Permafrost Carbon to CO₂.

- Geophys. Res. Lett.* **47**, 8 (2020a).
6. Bowen, J. C., Kaplan, L. A., Cory, R. M. Photodegradation disproportionately impacts biodegradation of semi-labile DOM in streams. *Limnol. Oceanogr.* **65**, 13-26 (2020b).
 7. Brandt, F. B., Breidenbach, B., Brenzinger, K., Conrad, R. Impact of short-term storage temperature on determination of microbial community composition and abundance in aerated forest soil and anoxic pond sediment samples. *Syst. Appl. Microbiol.* **37**, 570-577 (2014).
 8. Colman, B. P., Schimel, J. P. Drivers of microbial respiration and net N mineralization at the continental scale. *Soil Biol. Biochem.* **60**, 65-76 (2013).
 9. Cory, R. M., Crump, B. C., Dobkowski, J. A., Kling, G. W. Surface exposure to sunlight stimulates CO₂ release from permafrost soil carbon in the Arctic. *Proc. Natl. Acad. Sci. USA* **110**, 3429-3434 (2013).
 10. Cory, R. M., Ward, C. P., Crump, B. C., Kling, G. W. Sunlight controls water column processing of carbon in arctic fresh waters. *Science* **345**, 925-928 (2014).
 11. Cory, R. M., Kling, G. W. Interactions between sunlight and microorganisms influence dissolved organic matter degradation along the aquatic continuum. *Limnol. Oceanogr. Lett.* **3**, 102-116 (2018).
 12. Dean, J. F., Garnett, M. H., Spyrakos, E., Billett, M. F. The Potential Hidden Age of Dissolved Organic Carbon Exported by Peatland Streams. *J. Geophys. Res. Biogeosci.* **124**, 328-341 (2019).
 13. Dee, D. P. et al. The ERA-Interim reanalysis: configuration and performance of the data assimilation system. *Q. J. R. Meteorol. Soc.* **137**, 553-597 (2011).
 14. Delavaux, C. S., Bever, J. D., Karppinen, E. M., Bainard, L. D. Keeping it cool: Soil sample cold pack storage and DNA shipment up to 1 month does not impact metabarcoding results. *Ecol. Evol.* **10**, 4652-4664 (2020).
 15. Ding, J. et al. The permafrost carbon inventory on the Tibetan Plateau: a new evaluation using deep sediment cores. *Glob. Chang. Biol.* **22**, 2688-2701 (2016).
 16. Elberling, B. et al. Long-term CO₂ production following permafrost thaw. *Nat. Clim. Chang.* **3**, 890-894 (2013).
 17. Feng, W. et al. Enhanced decomposition of stable soil organic carbon and microbial catabolic potentials by long-term field warming. *Glob. Chang. Biol.* **23**, 4765-4776 (2017).
 18. Fontaine, S., Bardoux, G., Abbadie, L., Mariotti, A. Carbon input to soil may decrease soil carbon content. *Ecol. Lett.* **7**, 314-320 (2004).
 19. Fontaine, S., Barot, S., Barre, P., Bdioui, N., Mary, B., Rumpel, C. Stability of organic carbon in deep soil layers controlled by fresh carbon supply. *Nature* **450**, 277-280 (2007).
 20. Grossart, H. P., Van den Wyngaert, S., Kagami, M., Wurzbacher, C., Cunliffe, M., Rojas-Jimenz, K. Fungi in aquatic ecosystems. *Nat. Rev. Microbiol.* **17**, 339-354 (2019).
 21. Gueymard, C. A., Habte, A., Sengupta, M. Reducing Uncertainties in Large-Scale Solar Resource Data: The Impact of Aerosols. *IEEE J. Photovolt.* **8**, 1732-1737 (2018).

22. Gueymard, C. A. *Solar Resources Mapping* Ch. 5 (Springer, Cham, Switzerland, 2019).
23. Hammes, F., Vital, M., Egli, T. Critical Evaluation of the Volumetric “Bottle Effect” on Microbial Batch Growth. *Appl. Environ. Microbiol.* **76**, 1278-1281 (2010).
24. Hartzog, P. E., Sladek, M., Kelly, J. J., Larkin, D. J. Bottle effects alter taxonomic composition of wetland soil bacterial communities during the denitrification enzyme activity assay. *Soil Biol. Biochem.* **110**, 87-94 (2017).
25. Jia, J., Feng, X., He, J. S., He, H., Lin, L., Liu, Z. Comparing microbial carbon sequestration and priming in the subsoil versus topsoil of a Qinghai-Tibetan alpine grassland. *Soil Biol. Biochem.* **104**, 141-151 (2017).
26. Li, J. et al. Depth dependence of soil carbon temperature sensitivity across Tibetan permafrost regions. *Soil Biol. Biochem.* **126**, 82-90 (2018).
27. Liu, Y. et al. Regional variation in the temperature sensitivity of soil organic matter decomposition in China’s forests and grasslands. *Glob. Chang. Biol.* **23**, 3393-3402 (2017).
28. Louca, S., Parfrey, L. W., Doebeli, M. Decoupling function and taxonomy in the global ocean microbiome. *Science* **353**, 1272-1277 (2016).
29. Ma, Q., Jin, H., Yu, C., Bense, V. F. Dissolved organic carbon in permafrost regions: A review. *Sci. China Earth Sci.* **62**, 349-364 (2019).
30. Morrissey, E. M. et al. Bacterial carbon use plasticity, phylogenetic diversity and the priming of soil organic matter. *The ISME Journal* **11**, 1890-1899 (2017).
31. Nalven, S. G. et al. Experimental metatranscriptomics reveals the costs and benefits of dissolved organic matter photo-alteration for freshwater microbes. *Environ. Microbiol.* **22**, 3505-3521 (2020).
32. Nguyen, N. H. et al. FUNGuild: An open annotation tool for parsing fungal community datasets by ecological guild. *Fungal Ecol.* **20**, 241-248 (2016).
33. Pascault, N. et al. Stimulation of Different Functional Groups of Bacteria by Various Plant Residues as a Driver of Soil Priming Effect. *Ecosystems* **16**, 810-822 (2013).
34. Pound, H. L. et al. Changes in Microbiome Activity and Sporadic Viral Infection Help Explain Observed Variability in Microcosm Studies. *Front. Microbiol.* **13**, 809989 (2022).
35. Razanamalala, K. et al. Soil microbial diversity drives the priming effect along climate gradients: a case study in Madagascar. *The ISME Journal* **12**, 451-462 (2018).
36. Rijkers, R., Rousk, J., Aerts, R., Sigurdsson, B. D., Weedon, J. T. Optimal growth temperature of Arctic soil bacterial communities increases under experimental warming. *Glob. Chang. Biol.* **28**, 6050-6064 (2022).
37. Spohn, M., Klaus, K., Wanek, W., Richter, A. Microbial carbon use efficiency and biomass turnover times depending on soil depth – Implications for carbon cycling. *Soil Biol. Biochem.* **96**, 74-81 (2016).
38. Su, J. et al. Low carbon availability in paleosols nonlinearly attenuates temperature sensitivity of soil organic matter decomposition. *Glob. Chang. Biol.*

- 28, 4180-4193 (2022).
39. Tang, W. Dataset of high-resolution (3 hour, 10 km) global surface solar radiation (1983-2018). National Tibetan Plateau/Third Pole Environment Data Center <https://doi.org/10.11888/Meteoro.tpdc.270112> (2019).
 40. Tranvik, L. J., Bertilsson, S. Contrasting effects of solar UV radiation on dissolved organic sources for bacterial growth. *Ecol. Lett.* **4**, 458-463 (2001).
 41. Treat, C. C., Wollheim, W. M., Varner, R. K., Grandy, A. S., Talbot, J., Frohling, S. Temperature and peat type control CO₂ and CH₄ production in Alaskan permafrost peats. *Glob. Chang. Biol.* **20**, 2674-2686 (2014).
 42. Vahatalo, A. V., Salkinoja-Salonen, M., Taalas, P., Salonen, K. Spectrum of the quantum yield for photochemical mineralization of dissolved organic carbon in a humic lake. *Limnol. Oceanogr.* **45**, 664-676 (2000).
 43. Vonk, J. E. et al. Reviews and syntheses: Effects of permafrost thaw on Arctic aquatic ecosystems. *Biogeosciences* **12**, 7129-7167 (2015a).
 44. Vonk, J. E. et al. Biodegradability of dissolved organic carbon in permafrost soils and aquatic systems: a meta-analysis. *Biogeosciences* **12**, 6915-6930 (2015b).
 45. Wagai, R., Kishimoto-Mo, A. W., Yonemura, S., Shirato, Y., Hiradate, S., Yagasaki, Y. Linking temperature sensitivity of soil organic matter decomposition to its molecular structure, accessibility, and microbial physiology. *Glob. Chang. Biol.* **19**, 1114-1125 (2013).
 46. Wan, W. et al. A comprehensive data set of lake surface water temperature over the Tibetan Plateau derived from MODIS LST products 2001–2015. *Sci. Data* **4**, 170095 (2017).
 47. Wang, K. Homogeneous grid dataset of Chinese land surface observation (surface solar radiation, surface wind speed, relative humidity and land surface evapotranspiration). National Tibetan Plateau/Third Pole Environment Data Center <https://doi.org/10.11888/Atmos.tpdc.272817> (2022).
 48. Wang, Q., Liu, S., Tian, P. Carbon quality and soil microbial property control the latitudinal pattern in temperature sensitivity of soil microbial respiration across Chinese forest ecosystems. *Glob. Chang. Biol.* **24**, 2841-2849 (2018).
 49. Ward, C. P., Nalven, S. G., Crump, B. C., Kling, G. W., Cory, R. M. Photochemical alteration of organic carbon draining permafrost soils shifts microbial metabolic pathways and stimulates respiration. *Nature Communications* **8**, 772 (2017).
 50. Wauthy, M. et al. Increasing dominance of terrigenous organic matter in circumpolar freshwaters due to permafrost thaw. *Limnol. Oceanogr. Lett.* **3**, 186-198 (2018).
 51. Wetzel, R. G., Hatcher, P. G., Bianchi, T. S. Natural photolysis by ultraviolet irradiance of recalcitrant dissolved organic matter to simple substrates for rapid bacterial metabolism. *Limnol. Oceanogr.* **40**, 1369-1380 (1995).
 52. Xu, M., Li, X., Kuyper, T. W., Xu, M., Li, X., Zhang, J. High microbial diversity stabilizes the responses of soil organic carbon decomposition to warming in the subsoil on the Tibetan Plateau. *Glob. Chang. Biol.* **27**, 2061-2075 (2021).

53. Yang, J. et al. Potential utilization of terrestrially derived dissolved organic matter by aquatic microbial communities in saline lakes. *Isme J.* **14**, 2313-2324 (2020).
54. Yang, K., He, J., Tang, W., Lu, H., Qin, J., Chen, Y., Li, X. China meteorological forcing dataset (1979-2018). National Tibetan Plateau/Third Pole Environment Data Center <https://doi.org/10.11888/AtmosphericPhysics.tpe.249369.file> (2019).
55. Zhang, K., Ni, Y., Liu, X., Chu, H. Microbes changed their carbon use strategy to regulate the priming effect in an 11-year nitrogen addition experiment in grassland. *Sci. Total Environ.* **727**, 138645 (2020).
56. Zhang, Q. et al. Nitrogen addition stimulates priming effect in a subtropical forest soil. *Soil Biol. Biochem.* **160**, 108339 (2021).
57. Zhou, L. et al. Decreasing diversity of rare bacterial subcommunities relates to dissolved organic matter along permafrost thawing gradients. *Environ. Int.* **134**, 105330 (2020).
58. Zhu, E. et al. Inactive and inefficient: Warming and drought effect on microbial carbon processing in alpine grassland at depth. *Glob. Chang. Biol.* **27**, 2241-2253 (2021).

Reviewer #1 (Remarks to the Author):

Review of revised ms "Photo-produced small aromatic compounds stimulate microbial degradation of dissolved organic carbon in thermokarst lakes"

The authors addressed all my comments. Their response to reviewer comments is one of the most careful and rigorous I have read in my many years as a reviewer. Their detailed, rigorous responses including additional supporting data and analysis provides much confidence in their work. It was a real pleasure to read! Thank you!

Specifically, I really appreciated the extensive work the authors did to calculate rates of light absorption by CDOM. I understand their hesitancy to include these data in the main text, but hopefully some of these data such as Fig. R1 could be in the supporting information (along with statements on uncertainty). Re: the uncertainty—it is very common to use models like NCAR TUV for UV photon fluxes, so it would be fine to state that in the supporting information. Including that work done by the authors would be very helpful for readers trying to build on the excellent work done by these authors.

In addition, the photos in Fig R3 were really helpful for a reader like me to get a visual assessment of how important photochemical processes are in these lakes! I encourage the authors to include Fig. R3 in the supporting information. In addition, I suggest the authors refer to Fig. R3 in the main text (such as in the discussion or conclusion section) with a simple, qualitative comparison of CDOM and photon fluxes in their system to similarly shallow and unshaded arctic lakes where photo degradation of DOC has already been determined to be quantitatively important to regional and global carbon budgets. For example, the authors could compare their average or range of CDOM concentrations (a_{300} values from their Supplementary Table 3) to the average or range of CDOM in arctic lakes (e.g., Cory et al. 2014 supporting information figure S9, table S2) and then also compare the summer time (e.g., noontime on summer solstice) UV photon flux reaching lakes in these two regions (they have this data in their rebuttal file, and could be compared to arctic lakes very quickly using NCAR TUV, or compare to existing data in Figure S5 of Cory et al. 2014). Given that the CDOM is similar between the Tibetan Plateau thermokarst lakes and arctic lakes, and given that the UV photon flux is higher in the Tibetan Plateau thermokarst lakes than in arctic lakes (for clear sky conditions it must be because Tibetan Plateau thermokarst lakes are at lower latitude than arctic lakes), then photodegradation of DOC in these lakes almost certainly is important based on first principles of photochemistry (as reviewed in Cory & Kling 2018). It would also be very helpful for readers to know what the range of CDOM is (at 305 nm or somewhere in the UV) in the main text.

Reviewer #2 (Remarks to the Author):

This paper presents a study of the combined controls of sunlight exposure and microbial degradation on dissolved organic carbon mineralization in lakes formed by thermokarst failures of permafrost. These lakes vary in DOC content and serve as important conduits for the flux of carbon from land to atmosphere. The authors conducted incubation experiments in which they photo-exposed filter-sterilized DOC from a range of lakes with the equivalent of ~ one day of sunlight, and then inoculated that organic matter with bacteria from its lake of origin. They found many remarkably strong relationships between DOC chemistry and the effect of sunlight exposure on the degree to which sunlight enhanced or reduced microbial respiration. They concluded that photo-production of small aromatic compounds was the main mechanism by which sunlight enhanced microbial respiration. They also concluded that aromatic-compound degrading microbial taxa were abundant in their samples.

I think this was a well-conceived, high-quality study that found very strong relationships that are potentially ground-breaking. The methodology is sound, and the methods provide adequate detail. This work will be significant for the field.

With this revision, the authors addressed all of my concerns adequately and I think the manuscript

is in good shape for publication.

Responses to Reviewer #1

[Comment 1] The authors addressed all my comments. Their response to reviewer comments is one of the most careful and rigorous I have read in my many years as a reviewer. Their detailed, rigorous responses including additional supporting data and analysis provides much confidence in their work. It was a real pleasure to read! Thank you!

[Response] We are very grateful to the reviewer for the positive and insightful comments on our manuscript!

[Comment 2] Specifically, I really appreciated the extensive work the authors did to calculate rates of light absorption by CDOM. I understand their hesitancy to include these data in the main text, but hopefully some of these data such as Fig. R1 could be in the supporting information (along with statements on uncertainty). Re: the uncertainty—it is very common to use models like NCAR TUV for UV photon fluxes, so it would be fine to state that in the supporting information. Including that work done by the authors would be very helpful for readers trying to build on the excellent work done by these authors.

[Response] Following the reviewer's comments, we have added the results of rates of light absorption by CDOM and the statements on its uncertainty in the main text as follows: *“Furthermore, compared to the shallow arctic lakes (Cory et al., 2014), the photon flux in the thermokarst lakes on the Tibetan Plateau is generally higher (Supplementary Fig. 9), despite the similar content of CDOM in the two regions (a_{305} : 7.37~60.57 in this study vs 1.05~77.46 m^{-1} for arctic lakes)(Supplementary Table 3 and (Cory et al., 2014)). Given the rate of light absorption by CDOM ($Q_{a,\lambda}$) depends on the both light available and the content of CDOM, the high photon flux infers the high rates of photochemical process on the Tibetan Plateau. Specifically, the rate of light absorption by CDOM was estimated to range from 95.10 to 153.21 $\mu\text{mol photon } m^{-2} s^{-1}$ in the 280-400 nm across the 10 lakes (Supplementary Fig. 10; see Supplementary Methods for details), inferring that DOC photodegradation could be an important*

source of CO₂ from these thermokarst lakes across the Tibetan Plateau” (Page 14, line 293-304) and in the supporting information as follows: “The rate of light absorption by CDOM ($Q_{a,\lambda}$) is a product of photon dose and the concentration of CDOM available to absorb the light. We calculated $Q_{a,\lambda}$ in the 10 thermokarst lakes (Supplementary Figure 10) in 2020 as follows (Bowen et al., 2020):

$$Q_{a,\lambda} = \int_{\lambda_{\min}}^{\lambda_{\max}} E_{\lambda} \left(1 - e^{-a_{\text{CDOM},\lambda} \times z}\right) \frac{a_{\text{CDOM},\lambda}}{a_{\text{tot},\lambda}} d\lambda \quad (1)$$

where λ_{\min} and λ_{\max} are the minimum and maximum wavelengths of UV light absorbed by CDOM (280 nm and 400 nm, respectively). E_{λ} is the photon flux spectrum (mol photon $\text{m}^{-2} \text{nm}^{-1}$), which was predicted by the NREL SMARTS model (Simple Model of the Atmospheric Radiative Transfer of Sunshine, V.2.9.5) (Cory et al., 2014). Specifically, since there were no meteorological stations near these 10 thermokarst lakes, we used the reanalysis data from satellites as the input data (e.g., atmospheric pressure, relative humidity, ozone abundance, and aerosol optical depth) for the model (Supplementary Table 5). Nevertheless, due to the strong elevation effects on aerosols and water vapor (Gueymard et al., 2018; Gueymard, 2019), these reanalysis data would lead to certain uncertainties in estimating E_{λ} . The fraction of light absorbed by CDOM relative to other light-absorbing constituents, $a_{\text{CDOM},\lambda}/a_{\text{tot},\lambda}$, was often equal to 1 at all wavelengths (Cory et al., 2014). We assumed that the path length of light (z) was equivalent to the average depth of the lakes (ranging from 0.24 m to 0.68 m) in the field. In addition, to compare the E_{λ} between the thermokarst lakes on the Tibetan Plateau and the arctic lakes, the direct and diffuse photon fluxes of the Tibetan Plateau thermokarst lakes at noon (1300 hr) on the summer solstice (June 21) in 2020 (wavelength: 300 to 600 nm) (Supplementary Figure 9) were also predicted by the model” (Page 2, line 26-48).

Fig. R1. Sunlight absorbed by CDOM ($Q_{a,\lambda}$) in 10 thermokarst lakes. Different colors represent different lakes. The data in parentheses represent the integral value of sunlight absorbed by CDOM from 280 to 400 nm for each lake ($\mu\text{mol photons m}^{-2} \text{s}^{-1}$). The ten sampling sites are Yushu (YS), Golmud (GM), Nagqu (NQ), Amdo County (AD), Nyainrong County (NR), Seni District (SN), Heihe River (HH), Golog Tibetan Autonomous Prefecture (GL), Madoi County (MD), and Maqên County (MQ).

Table R1. List of reanalysis data from satellites as SMARTS model input data and their sources.

Variable	Publication	Data sources
SPR (mb)	Yang et al. (2019)	https://data.tdpc.ac.cn/zh-hans/data/8028b944-daaa-4511-8769-965612652c49/
RH (%)	Wang (2022)	https://www.tpdc.ac.cn/zh-hans/data/99dd84e2-288d-4db8-b098-8118b3b0c17a
TDAY (°C)	National Earth System Science Data Center, National Science & Technology Infrastructure of China	http://www.geodata.cn/datapplication/OrderStepList.html?dataguid=250085273409240
IH2O (cm)	Dee et al. (2011)	https://doi.org/10.1002/qj.828
TAU550	National Earth System Science Data Center, National Science & Technology Infrastructure of China	http://www.geodata.cn/data/datadetails.html?dataguid=1776940&docId=3316

Notes: SPR, surface pressure; RH, relative humidity; TDAY, the average daily temperature at the site level; IH2O (cm), precipitable water; TAU550, aerosol optical depth at 550 nm, τ_{550} .

[Comment 3] In addition, the photos in Fig R3 were really helpful for a reader like me to get a visual assessment of how important photochemical processes are in these lakes! I encourage the authors to include Fig. R3 in the supporting information. In addition, I suggest the authors refer to Fig. R3 in the main text (such as in the discussion or conclusion section) with a simple, qualitative comparison of CDOM and photon fluxes in their system to similarly shallow and unshaded arctic lakes where photo degradation of DOC has already been determined to be quantitatively important to regional and global carbon budgets. For example, the authors could compare their average or range of CDOM concentrations (a_{300} values from their Supplementary Table 3) to the average or range of CDOM in arctic lakes (e.g., Cory et al. 2014 supporting information figure S9, table S2) and then also compare the summer time (e.g., noontime on summer solstice) UV photon flux reaching lakes in these two regions (they have this data in their rebuttal file, and could be compared to arctic lakes very quickly using NCAR TUV, or compare to existing data in Figure S5 of Cory et al. 2014). Given that the CDOM is similar between the Tibetan Plateau thermokarst lakes and arctic lakes, and given that the UV photon flux is higher in the Tibetan Plateau thermokarst lakes than in arctic lakes (for clear sky conditions it must be because Tibetan Plateau thermokarst lakes are at lower latitude than arctic lakes), then photodegradation of DOC in these lakes almost certainly is important based on first principles of photochemistry (as reviewed in Cory & Kling 2018). It would also be very helpful for readers to know what the range of CDOM is (at 305 nm or somewhere in the UV) in the main text.

[Response] Following the reviewer's comments, we have included Fig. R2 in the supporting information and added the relevant description in the revised MS as follows: *"In addition, these thermokarst lakes are generally located in swamp meadows and alpine meadows with limited vegetation growing in the lakes (Supplementary Fig. 11). Therefore, no vegetation shading will affect the importance of photo-bio degradation in these lakes"* (Page 15, line 327-330). In addition, we also compared the CDOM contents and photon fluxes between thermokarst lakes on the Tibetan Plateau and Arctic

lakes in the revised MS as follows: “Furthermore, compared to the shallow arctic lakes (Cory et al., 2014), the photon flux in the thermokarst lakes on the Tibetan Plateau is generally higher (Supplementary Fig. 9), despite the similar content of CDOM in the two regions (range of a_{305} : 7.37~60.57 vs 1.05~77.46 m^{-1} ; calculated from Supplementary Table 3 and (Cory et al., 2014)). Given the rate of light absorption by CDOM ($Q_{a,\lambda}$) depends on the both light available and the content of CDOM, the high photon flux infers the high rates of photochemical process on the Tibetan Plateau. Specifically, the rate of light absorption by CDOM was estimated to range from 95.10 to 153.21 $\mu\text{mol photon m}^{-2} \text{s}^{-1}$ in the 280-400 nm across the 10 lakes (Supplementary Fig. 10; please refer to the supplementary material for detail calculation and uncertainty analysis) (Cory et al., 2018), inferring that DOC photodegradation could be an important source of CO_2 from these thermokarst lakes across the Tibetan Plateau” (Page 14, line 293-304).

Fig. R2. Aerial and close-up images of thermokarst lakes on the Tibetan plateau. Aerial images (left) and close-up images (right) of thermokarst lakes in Madoi County (MD, a) and Maqên County (MQ, b) in swamp meadows and alpine meadows. The

photographs were captured by Ziliang Li and Luyao Kang in August 2020.

Fig. R3. Direct and diffuse (sky) photon flux spectrum (E_{λ}) above the 10 Tibetan Plateau thermokarst lakes water surface at 1300 hr local time on 21 June 2020. Solid lines represent mean curves, shaded areas represent means \pm standard deviation ($n = 10$). The pink and blue lines indicate direct and diffuse photon flux densities respectively.

Table R2. Climate and water properties of 10 thermokarst lakes on the Tibetan alpine permafrost region.

Site	MAP (mm)	MAT (°C)	Salinity (ppt)	Conductivity (mS cm ⁻¹)	DO (mg L ⁻¹)	DOC (mg L ⁻¹)	DIC (mg L ⁻¹)	DON (mg L ⁻¹)	pH	a ₃₀₀ (m ⁻¹)	a ₃₀₅ (m ⁻¹)	SUVA ₂₅₄			
												S _R	BIX	HIX	
YS	306.90	-4.03	0.36	606.67	8.20	6.62	17.64	0.41	8.33	8.44	7.37	1.40	1.29	0.71	0.81
GM	343.60	-2.77	0.25	416.03	4.48	7.37	18.96	0.41	8.34	18.96	17.20	2.35	0.92	0.59	0.88
NQ	444.93	-1.80	0.09	165.13	13.89	8.48	7.53	0.45	8.01	16.89	15.28	1.98	1.19	0.67	0.86
SN	464.90	-0.13	0.25	444.90	6.51	16.29	16.57	0.63	8.22	65.87	60.57	3.45	0.83	0.51	0.94
NR	464.00	-0.83	0.08	140.60	10.15	10.88	7.02	0.61	7.85	32.86	29.86	2.75	0.92	0.61	0.88
AD	456.53	-0.77	0.30	463.70	6.47	11.66	24.04	0.56	8.06	40.65	44.14	3.49	0.95	0.57	0.90
GL	409.03	-1.03	0.21	314.93	4.29	12.16	14.43	1.13	8.06	34.70	31.24	2.54	0.89	0.60	0.92
MD	436.73	-3.17	0.11	169.33	8.10	13.44	9.13	0.81	7.94	39.00	35.39	2.70	0.93	0.55	0.91
HH	373.97	-2.97	0.12	206.77	8.30	5.69	17.48	0.33	8.14	13.74	12.28	2.27	1.03	0.64	0.88
MQ	496.63	-0.87	0.10	147.63	6.56	16.76	7.59	1.35	7.83	54.89	49.90	3.08	0.89	0.56	0.92

Notes: MAP, mean annual precipitation; MAT, mean annual temperature; DO, dissolved oxygen; DOC, dissolved organic carbon; DIC, dissolved inorganic carbon; DON, dissolved organic nitrogen; a₃₀₀ and a₃₀₅, the Napierian absorption coefficient indicating chromophoric DOC content at 300 nm and 305nm, respectively; SUVA₂₅₄, a proxy for DOM aromaticity expressed as the absorbance at 254 nm divided by DOC concentration; S_R, slope coefficient ratio correlated to DOM molecular weight; BIX, biological index; HIX, humification index.

Overall, we greatly appreciate the reviewer's insightful comments. As mentioned above, we have included the comparison of the CDOM contents and photon fluxes between thermokarst lakes on the Tibetan Plateau and Arctic lakes in the main text. Accordingly, the results of the rates of light absorption by CDOM and its uncertainty were also included in the supplementary information section. By doing so, we feel that our revised manuscript has been greatly improved and expect that the reviewer will be satisfied with the revised manuscript. Thank you!

Responses to Reviewer #2

[Comment] This paper presents a study of the combined controls of sunlight exposure and microbial degradation on dissolved organic carbon mineralization in lakes formed by thermokarst failures of permafrost. These lakes vary in DOC content and serve as important conduits for the flux of carbon from land to atmosphere. The authors conducted incubation experiments in which they photo-exposed filter-sterilized DOC from a range of lakes with the equivalent of ~ one day of sunlight, and then inoculated that organic matter with bacteria from its lake of origin. They found many remarkably strong relationships between DOC chemistry and the effect of sunlight exposure on the degree to which sunlight enhanced or reduced microbial respiration. They concluded that photo-production of small aromatic compounds was the main mechanism by which sunlight enhanced microbial respiration. They also concluded that aromatic-compound degrading microbial taxa were abundant in their samples. I think this was a well-conceived, high-quality study that found very strong relationships that are potentially ground-breaking. The methodology is sound, and the methods provide adequate detail. This work will be significant for the field. With this revision, the authors addressed all of my concerns adequately and I think the manuscript is in good shape for publication.

[Response] Thanks for the reviewer's positive comments and recognition for our revision.

References

1. Bowen, J. C., Kaplan, L. A., Cory, R. M. Photodegradation disproportionately impacts biodegradation of semi-labile DOM in streams. *Limnol. Oceanogr.* **65**, 13-26 (2020).
2. Cory, R. M., Ward, C. P., Crump, B. C., Kling, G. W. Sunlight controls water column processing of carbon in arctic fresh waters. *Science* **345**, 925-928 (2014).
3. Cory, R. M., Kling, G. W. Interactions between sunlight and microorganisms influence dissolved organic matter degradation along the aquatic continuum. *Limnol. Oceanogr. Lett.* **3**, 102-116 (2018).

4. Dee, D. P. et al. The ERA-Interim reanalysis: configuration and performance of the data assimilation system. *Q. J. R. Meteorol. Soc.* **137**, 553-597 (2011).
5. Gueymard, C. A., Habte, A., Sengupta, M. Reducing Uncertainties in Large-Scale Solar Resource Data: The Impact of Aerosols. *IEEE J. Photovolt.* **8**, 1732-1737 (2018).
6. Gueymard, C. A. *Solar Resources Mapping* Ch. 5 (Springer, Cham, Switzerland, 2019).
7. Wang, K. Homogeneous grid dataset of Chinese land surface observation (surface solar radiation, surface wind speed, relative humidity and land surface evapotranspiration). National Tibetan Plateau/Third Pole Environment Data Center <https://doi.org/10.11888/Atmos.tpdc.272817> (2022).
8. Yang, K., He, J., Tang, W., Lu, H., Qin, J., Chen, Y., Li, X. China meteorological forcing dataset (1979-2018). National Tibetan Plateau/Third Pole Environment Data Center <https://doi.org/10.11888/AtmosphericPhysics.tpe.249369.file> (2019).